# M2 macrophages independently promote beige adipogenesis via blocking adipocyte Ets1

Suyang Wu [1,5], Chen Qiu[1,2,3,5], Jiahao Ni[1], Wenli Guo[1], Jiyuan Song[1], Xingyin Yang[1], Yulin Sun[1], Yanjun Chen[1], Yunxia Zhu [1], Xiaoai Chang[1], Peng Sun [1], Chunxia Wang[4], Kai Li [1,2] ✉ & Xiao Han [1] ✉

Adipose tissue macrophages can promote beige adipose thermogenesis by altering local sympathetic activity. Here, we perform sympathectomy in mice and further eradicate subcutaneous adipose macrophages and discover that these macrophages have a direct beige-promoting function that is independent of sympathetic system. We further identify adipocyte Ets1 as a vital mediator in this process. The anti-inflammatory M2 macrophages suppress Ets1 expression in adipocytes, transcriptionally activate mitochondrial biogenesis, as well as suppress mitochondrial clearance, thereby increasing the mitochondrial numbers and promoting the beiging process. Male adipocyte Ets1 knock-in mice are completely cold intolerant, whereas male mice lacking Ets1 in adipocytes show enhanced energy expenditure and are resistant to metabolic disorders caused by high-fat-diet. Our findings elucidate a direct communication between M2 macrophages and adipocytes, and uncover a function for Ets1 in responding to macrophages and negatively governing mitochondrial content and beige adipocyte formation.

Thermogenic (brown and beige) adipocytes are characterized by their unique ability to burn fat by uncoupling mitochondrial respiration from ATP synthesis. Brown adipocytes were first described in rodents and human infants[1–3]; however, the size and activity of brown adipose tissue are reduced in adults, especially in those individuals with obesity and type 2 diabetes[4,5]. Recent single cell sequencing (sc-seq) has revealed that beige adipocytes, induced by cold stress and physical exercise, originate from specific progenitor subsets within subcutaneous fat[6–9]. Considering the huge body surface area, exploiting this "beiging" process represents a promising approach for increasing energy expenditures and ameliorating obesity-related metabolic diseases[10].

Beige thermogenesis is regulated by both the sympathetic nervous system (SNS) and adipose tissue macrophages (ATMs), but the signals that initiate and sustain this process have not been fully clarified. The sympathetic fibers of subcutaneous fat originate from the celiac ganglia, which are activated by cold challenge[11]. Once activated, these fibers release norepinephrine (NE)[12], which then binds to the adipocyte β3 adrenergic receptor (AR) and activates thermogenesis by the cyclic AMP/protein kinase A (cAMP/PKA) pathway[13]. However, a sympathectomized rodent is completely resistant to cold stress[14], indicating that the NE secreted from the SNS is not solely responsible for changes in thermogenic fat activity[15–17].

[1]Key Laboratory of Human Functional Genomics of Jiangsu Province, Department of Biochemistry and Molecular Biology, Nanjing Medical University, Nanjing 211166, China. [2]Department of Endocrinology, The Affiliated Taizhou People's Hospital of Nanjing Medical University, Taizhou School of Clinical Medicine, Nanjing Medical University, Taizhou 225300, China. [3]Key Laboratory of the Model Animal Research, Animal Core Facility of Nanjing Medical University, Nanjing 211166, China. [4]Laboratory of Critical Care Translational Medicine, Institute of Pediatric Infection, Immunity, and Critical Care Medicine, Shanghai Jiao Tong University School of Medicine, Shanghai 200062, China. [5]These authors contributed equally: Suyang Wu, Chen Qiu. ✉e-mail: likai87@njmu.edu.cn; hanxiao@njmu.edu.cn

Adipose tissue becomes neuropathic (i.e., a reduction in neurites) in various conditions of metabolic dysregulation[18], and the involvement of ATM is presently discussed as being an integral part of beige adipogenesis[19]. Convergent studies have shown that M2-like macrophages regulate beiging via effects on the SNS, as deletion of Irs2[20], Mecp2[21], or a secretory factor slit3[22] in macrophages alters local sympathetic activity and thus affects the beiging process. Interestingly, separate studies have demonstrated the possibility that macrophages directly regulate beige adipogenesis either by eliminating NE[23,24] or secreting NE[25,26]. However, the suggestion that macrophages themselves can synthesize and secrete NE was questioned in later research[27]. Leaving aside the controversial NE, several other cytokines, such as IL-10[28], IL-1β[29] and TGFβ[30], that are secreted by macrophages and other immune cells reportedly target adipose stem and progenitor cells (ASPC) and regulate beige adipogenesis. These observations raise the possibility that ATMs might directly promote beige adipogenesis, in addition to the indirect effect mediated by the SNS. If so, the unanswered question is how ASPCs/beige adipocytes respond to ATMs and switch on thermogenesis?

In the present study, we performed a 6-hydroxydopamine (6OHDA)-mediated sympathectomy on mice and determined that eradicating macrophages also inhibited thermogenesis in these denervated mice. In other words, macrophages appeared to have a beige-promoting function that was independent of the SNS. Multiple sequence-based screening identified Ets1, a transcription factor that plays a fundamental role in maintaining glucolipid homeostasis[31,32], was a mediator of M2 macrophage stimuli. Ets1 expression by adipocytes suppressed mitochondrial biogenesis and activated mitochondrial clearance via classic transcriptional regulation. M2 macrophages could effectively inhibit the expression of Ets1 in adipocytes, thereby directly increasing the mitochondrial numbers and promoting the adipocyte beiging process.

## Results

### Macrophages directly regulate the adipocyte beiging process independent of the sympathetic nervous system

We eliminated the influence of the SNS on beige adipogenesis by conducting sympathectomy on wild type C57 mice by intraperitoneal injection of 6OHDA. After the 6OHDA ablation, the NE level in subcutaneous adipose tissue of the denervated mice decreased to a mere 5% of the control value (SuppFig. 1A). Fluorescence staining of tyrosine hydroxylase (TH), a SNS marker, revealed the overall reduction of the sympathetic fiber within the fat pad (SuppFig. 1B, C).

To further determine whether macrophages alone played an essential role in beige adipogenesis, we employed the pharmacologic approach of Clodronate Liposomes (Clod Lipo) to locally ablate the macrophage in inguinal white adipose tissue (iWAT). The expression of F4/80, a marker gene of macrophages, was significantly reduced in iWAT after liposome injection (SuppFig. 1D). The expression of *F4/80* in epididymal adipose tissue (SuppFig. 1D), and the number of blood monocytes (SuppFig. 1E) were not affected by local injection of Clod Lipo, indicating that Clod Lipo is limited to iWAT, and not speech systemically. We then subjected these macrophage null/SNS denervated mice to 4 days of cold stress, which would normally stimulate adipose beiging. Neither the 6OHDA nor the Clod Lipo treatment affected the body weight (SuppFig. 1F) or iWAT tissue weight (SuppFig. 1G). Compared with control group without 6OHDA or clodronate liposome treatment, both the macrophage depletion and SNS ablation can significantly inhibit adipocyte beiging and thermogenesis, manifested as lower body temperature (Fig. 1A), less expression of thermogenic genes (Fig. 1B, C), and corresponding morphological changes after cold stimulation (Fig. 1D). Interestingly, comparing with the control SNS-denerved mice, the macrophage-null/SNS-denerved mice showed even lower thermogenic capacity

(Fig. 1A–D), indicating a surprising finding that macrophages alone could induce adipose beiging independent of SNS input.

### Multiple sequencing-based screening identifies Ets1 as a potential governor in macrophage-adipocyte communication

To explore the mechanism through which macrophage directly modulate adipocyte beiging, we set up an analysis pipeline (Fig. 1E). Considering the key role of transcriptional regulation in the differentiation of beige adipocytes[33], we focused on DNA binding proteins (DBPs), such as transcription factors and epigenetic regulators. The candidate DBPs were screened according to the following limits.

The DBPs is bio-active in adipocyte linage. We first performed the Assay for Transposase Accessible Chromatin with high-throughput sequencing (ATAC-seq) to profile the global open chromatin structure of the white and beige adipocytes. Comparison with a published FAIRE-seq dataset[34] revealed that our ATAC-seq had more reliable sequencing signals with much lower background (SuppFig. 1H). Motif analysis of the open chromatin regions, which can identify active and functional DBPs, revealed 30 active DBPs in white adipocytes and 50 in beige adipocytes, with 29 shared DBPs (Fig. 1F).

The adipocyte bio-active DBPs could response to M2 macrophage stimuli. Using a published RNA-seq dataset for macrophage co-cultured adipocytes (GSE228094), we found that compared with the mono-cultured adipocytes, the adipocytes co-cultured with M2 macrophages showed reduced expression of *Ets1*, and increased *Klf4* expression (Fig. 1G). Klf4 is considered an essential early regulator of adipocyte differentiation[35], but no clear physiological function has been established for Ets1 in adipocytes. Interestingly, unlike *Klf4*, adipocytes co-cultured with M1 macrophages (Fig. 1G), which promote inflammation and impede beige adipogenesis[36], showed increased *Ets1* expression. These results suggested that Ets1 could be a potential regulator of macrophage–adipocyte communication.

The adipocyte bio-active, M2 macrophage responsible DBP, should be highly associated with the beige adipogenesis process, either in its expression or in its activity. Using a published adipose tissue sc-sequencing database[8], we profiled the expression of *Ets1* in a variety of cells located in adipose tissue and found that mice fed a normal chow diet (NCD) showed *Ets1* expression mainly in ASPCs, B cells, macrophages, endothelial cells, and mesothelial cells (Fig. 1H). By contrast, only a very small portion of the mature adipocytes showed *Ets1* expression (Fig. 1H). Interestingly, mice fed a high-fat diet (HFD) had higher numbers of mature adipocytes that expressed *Ets1* (SuppFig. 1I). We then conducted a trajectory analysis of ASPCs and mature adipocytes to determine the developmental pathway (SuppFig. 1J, K). Differential expression analysis identified several gene cohorts that changed during this ASPCs-mature adipocytes transition. Notably, the transition was marked by downregulation of *Ets1*, together with other gene signatures found in (but not specific to) ASPCs, such as *Dpp4*, *Fstl1*[37], and *Egfr*[38], and upregulation of mature adipocyte marker genes, such as *Pparg*, *Pnpla3*, *Lpl*, *Adipoq*, and *Fabp4* (Fig. 1I). These data revealed a negative correlation between Ets1 and the differentiation process from ASPCs to adipocytes.

### Expression of Ets1 is negatively associated with adipose beiging process

To confirm the sequencing-based screening result, we collected conditioned medium of M2 macrophages (M2 CDM) and used to incubate them with 3T3-L1, fibroblast with high adipogenesis capacity. The M2 CDM dose-dependently inhibited the expression of Ets1 in 3T3-L1 (Fig. 2A), consistent with the RNA-seq data (Fig. 1G). The suppressing effect was further confirmed by adipose tissue macrophage CDM co-culturing assay (Fig. 2A). Separation of the primary stromal vascular fraction (SVF) cells and mature adipocytes by density gradient centrifugation revealed that Ets1 was mainly expressed in the ASPC-enriched SVF cells (Fig. 2B), consistent with the sc-seq data (Fig. 1H).

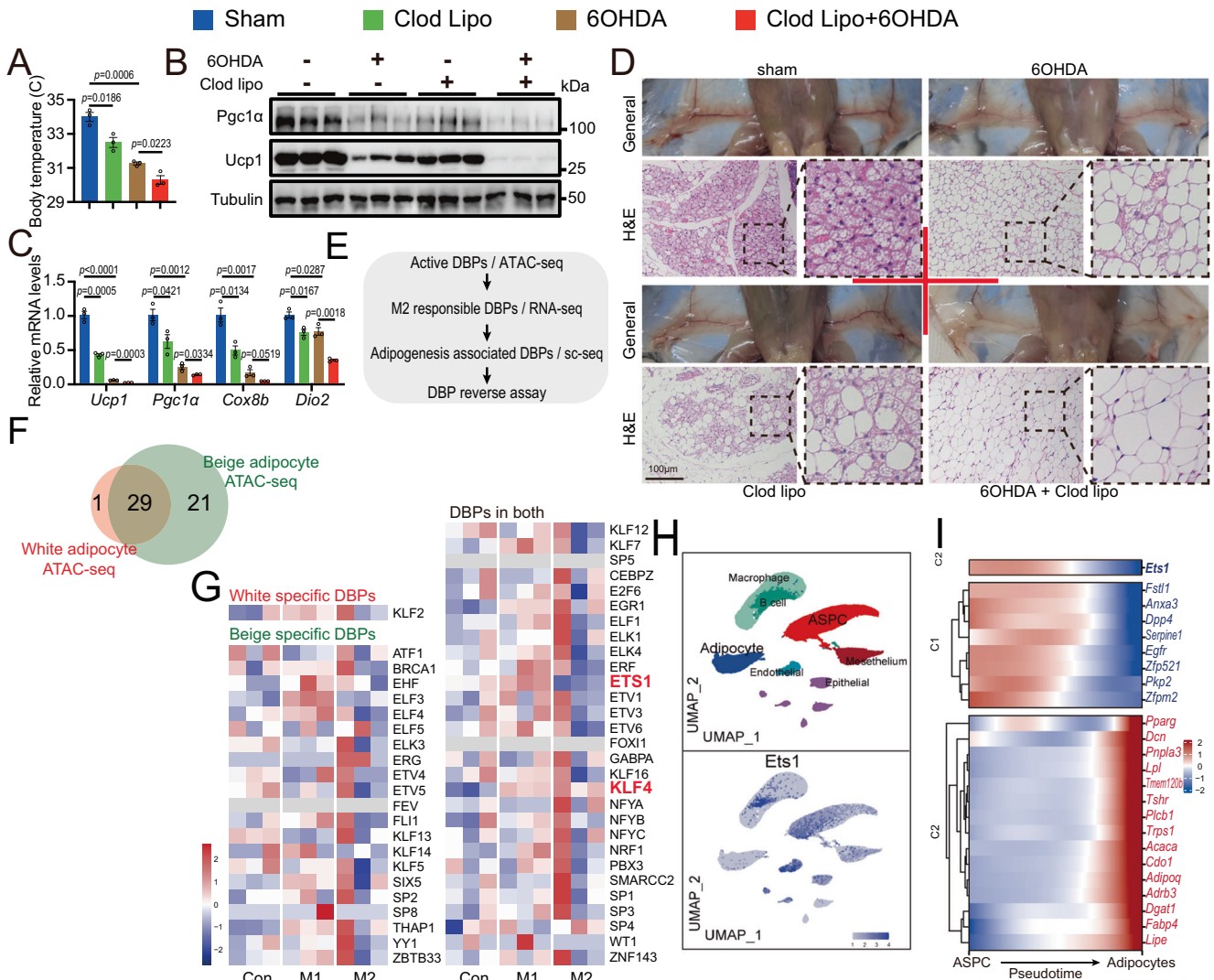

**Fig. 1 | Macrophages directly regulate the adipocyte beiging process independent of the sympathetic nervous system. A–D** All the four group of sham ($n = 3$), 6-Hydroxydopamine (6OHDA) alone ($n = 3$), Clodronate Liposomes (Clod Lipo) alone ($n = 3$), and 6OHDA+Clod Lipo ($n = 3$) were cold stressed for 4 days, started from Day 0. **A** Rectal temperature measured after a 4-day cold stress. **B** Western blot analyzing the protein level of thermogenic marker Pgc1α and Ucp1. The experiment was repeated two times independently. **C** Quantitative PCR analyzing the mRNA level of beige adipose marker genes. **D** General morphology of inguinal white adipose tissue (iWAT) (up) and Hematoxylin and eosin (H&E) staining of iWAT (down). Scale bar: 100 μm. **E–I** Multiple sequencing-based screening identifies Ets1 as a potential governor in M2 macrophage-adipocyte communication. **E** Analysis pipeline. **F** De novo motif analysis of the open peaks identified adipocyte-specific DNA binding proteins (DBPs). Stromal vascular fraction (SVF) separated from iWAT of wild type C57 mice was induced differentiation towards beige or white adipocyte for 7 days. The mature adipocytes were then analyzed by ATAC-seq. **G** The expression of the 51 uncovered DBPs were profiled, using a published RNA-seq dataset of M1 or M2 macrophage co-cultured adipocyte (GSE228094). **H** Analyzing the expression of Ets1 in adipose tissue, using a published single-cell transcription sequencing dataset (GSE176171). Uniform manifold approximation and projection (UMAP) and unsupervised clustering of chow diet fed mice adipose tissue were performed (up), the expression of Ets1 across these cells were profiled (down). **I** Pseudotime differential trajectory analysis of adipose stem and progenitor cells (ASPC) and mature adipocytes, inferred by the Slingshot method. Heatmap denoting genes that become differentially expressed during pseudotime ordering of ASPC-to- adipocyte transition. Ets1, as well as the known markers / enriched genes of mature adipocyte and ASPC were list. Data are means ± SEM. Two-sided Student's t-test was used to evaluate statistical significance. Source data are provided as a Source Data file.

We then analyzed the association between Ets1 expression and adipose beiging. Mice kept at 4 °C showed time-dependent reductions in both protein (Fig. 2C) and mRNA (SuppFig. 2A) expression of Ets1 in iWAT. This reduction could be mainly attributed to mature adipocytes, whereas the expression of Ets1 in SVF cells was unchanged (SuppFig. 2B). Consistent in vitro results were also obtained using SVF-derived beige adipocytes (Fig. 2D, SuppFig. 2C). Consistently, Ets1 expression was higher in adipocytes from *db/db* diabetic mice and HFD-fed mice—two well-known obesity models that show pro-inflammatory M1 macrophage polarization (Fig. 2E, SuppFig. 2D).

## M2 macrophage directly activates adipocytes thermogenesis via blocking Ets1

The regulatory effect of Ets1 on beige adipogenesis was then explored. Overexpression of Ets1 dramatically decreased the differentiation capacity of the beige adipocytes, as reflected by the reduced expression of thermogenic marker genes (Fig. 3A, B) and by oil red O staining (Fig. 3C). Functional beige adipocytes burn fat and generate heat by decoupling oxidative phosphorylation. Seahorse assays showed that the control cells had comparable oxygen consumption rates between stage 3 (maximum) and stage 1 (basal), suggesting that the mitochondria were already decoupled and the adipocytes were functional

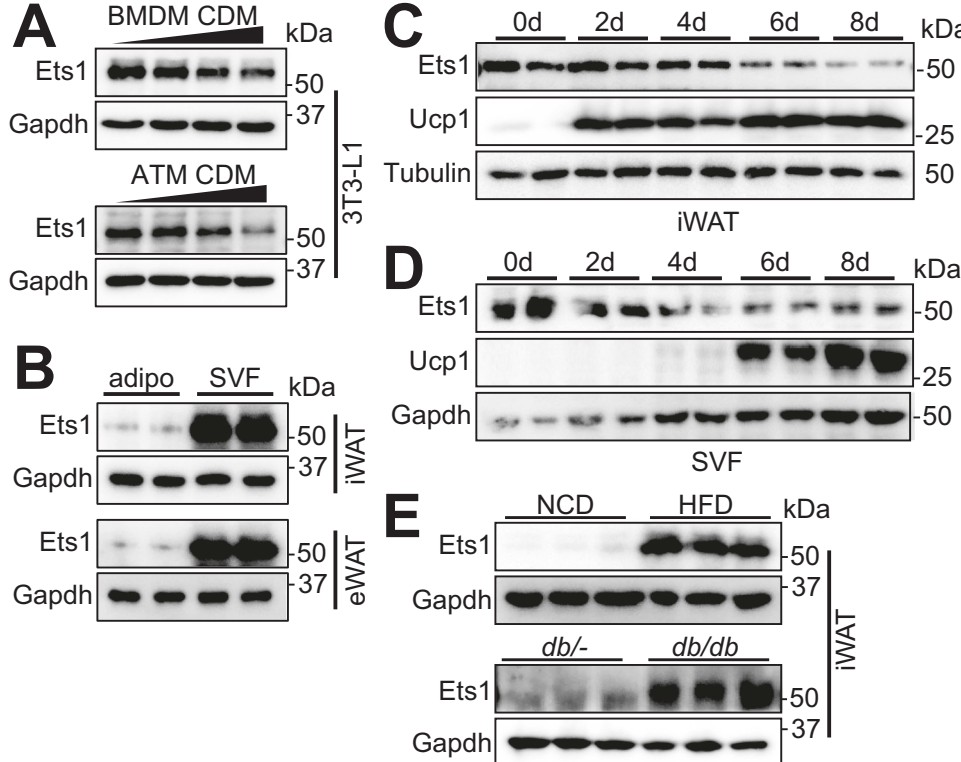

**Fig. 2 | Adipocyte Ets1 is negatively associated with beige adipogenesis. A** 3T3-L1 cells were incubated with 0%/20%/30%/50% conditional medium (CDM) for 24 h, followed by testing the expression of Ets1 via western blot. Up: CDM of M2 bone marrow-derived macrophages (BMDM) were used; Down: CDM of adipose tissue macrophage (ATM) of mice pretreated with 48 h cold stress, were used ($n = 2$). **B** Western blot of Ets1 in mature adipocytes (adipo) and in the SVF of iWAT and epididymal white adipose tissue (eWAT) ($n = 2$). **C** Western blot showing the protein levels of Ets1 in iWAT at 0, 2, 4, 6, and 8 days after cold stress ($n = 2$). **D** SVF separated from wild type C57 was differentiated into beige adipocytes. At indicated time point after differentiation, protein level of Ets1 were analyzed by western blot ($n = 2$). **E** Western blot of Ets1 in mature adipocyte derived from iWAT. Up: High-fat diet (HFD) and normal chow diet (NCD) feedings were started at 4 weeks of age and continued for 16 weeks. Down: *db/-* and *db/db* mice were 8–10 weeks old. $N = 3$ for each group. All blot assay was repeated two times independently.

thermogenic fat. Ets1 overexpression significantly blocked both the basal and the maximal respiration (Fig. 3D).

A reverse assay to determine whether M2 macrophages stimulate beige adipogenesis by blocking Ets1 expression revealed an increased expression of beige adipocyte marker genes after M2 CDM treatment (Fig. 3E, F). The M2 CDM treatment also raised the basal and maximum oxygen consumption rate (Fig. 3G), indicating better thermogenic function of the beige adipocytes. Notably, forced expression of Ets1 dramatically blocked these M2 CDM-induced changes (Fig. 3E–G).

Interestingly, adipocytes treated with M1 macrophages conditional medium (CDM) showed enhanced Ets1 expression (Supp-Fig. 2E), which is exactly opposite to the effect of M2 macrophages CDM. In addition, overexpression of Ets1 can significantly exacerbate the inhibitory effect of M1 macrophages on beige adipogenesis (SuppFig. 2E, F).

### Adipocyte-specific knock-in Ets1 impaired beige adipose thermogenesis

We then analyzed the regulatory effect of Ets1 on the beiging process by generating adipocyte-specific Ets1 knock-in mice (Ets1$^{ki/ki}$×Adipoq-cre, abbreviated as EA+/+). Western blotting confirmed that the enhanced expression of Ets1 was restricted only to the fat pads (SuppFig. 3A). Compared with their control littermates (Ets1$^{ki/ki}$, abbreviated as EC +/+), the EA+/+ mice showed comparable body weight (SuppFig. 3B), adipose tissue weight (SuppFig. 3C, D), and showed normal fertility and growth under normal feeding conditions at room temperature. However, the homozygous EA+/+ mice were completely intolerant of a 4 °C cold stress (Fig. 4A).

The heterozygous knockin mice (Ets1$^{ki/-}$×Adipoq-cre, abbreviated as EA+/-), together with their control counterparts (Ets1$^{ki/-}$, abbreviated as EC+/-), were used in a further cold stress assay. The rectal temperature was lower in the EA+/- mice than in the control group, indicating a reduced thermogenic capacity (Fig. 4B). Histological staining of the fat pads revealed an increase in the size of the lipid droplets in the iWAT of the EA+/- mice (Fig. 4C). Consistently, the expression of thermogenic marker genes (Fig. 4D, E) was reduced in the iWAT of the EA+/- mice. Interestingly, the forced expression of Ets1 showed only a minimal effect in the brown adipose tissue (BAT) (Fig. 4C–E).

The negative effects of Ets1 on the adipocyte beiging process were further analyzed using Cl316243, a β3 AR agonist. Treatment of homozygous mice gave results generally similar (Fig. 4F–H) to those obtained in the heterozygous mice exposed to a cold stress. However, a notable finding was that the diverse expression of Ets1 in the EA+/- and EA+/+ mice was strongly associated with the level of repression of thermogenic fat activation. The EA+/- mice showed approximately 4-fold increases in the Ets1 mRNA levels in iWAT and approximately 8-fold increases in BAT; while the EA+/+ mice showed approximately 9-fold increases Ets1 expression in iWAT and approximately 15-fold increases in BAT (Fig. 4H). The higher Ets1 level in the EA+/+ mice led to more severe histological changes in iWAT (Fig. 4F), together with a dramatic reduction in the expression of Upc1, Pgc1α (Fig. 4G), and other thermogenic genes (Fig. 4H). Consistently, the effects of Ets1 overexpression on BAT thermogenic activation were relatively small (Fig. 4F–H).

Thermogenic fat activation leads to increases in mitochondria numbers, thereby increasing oxygen consumption and energy

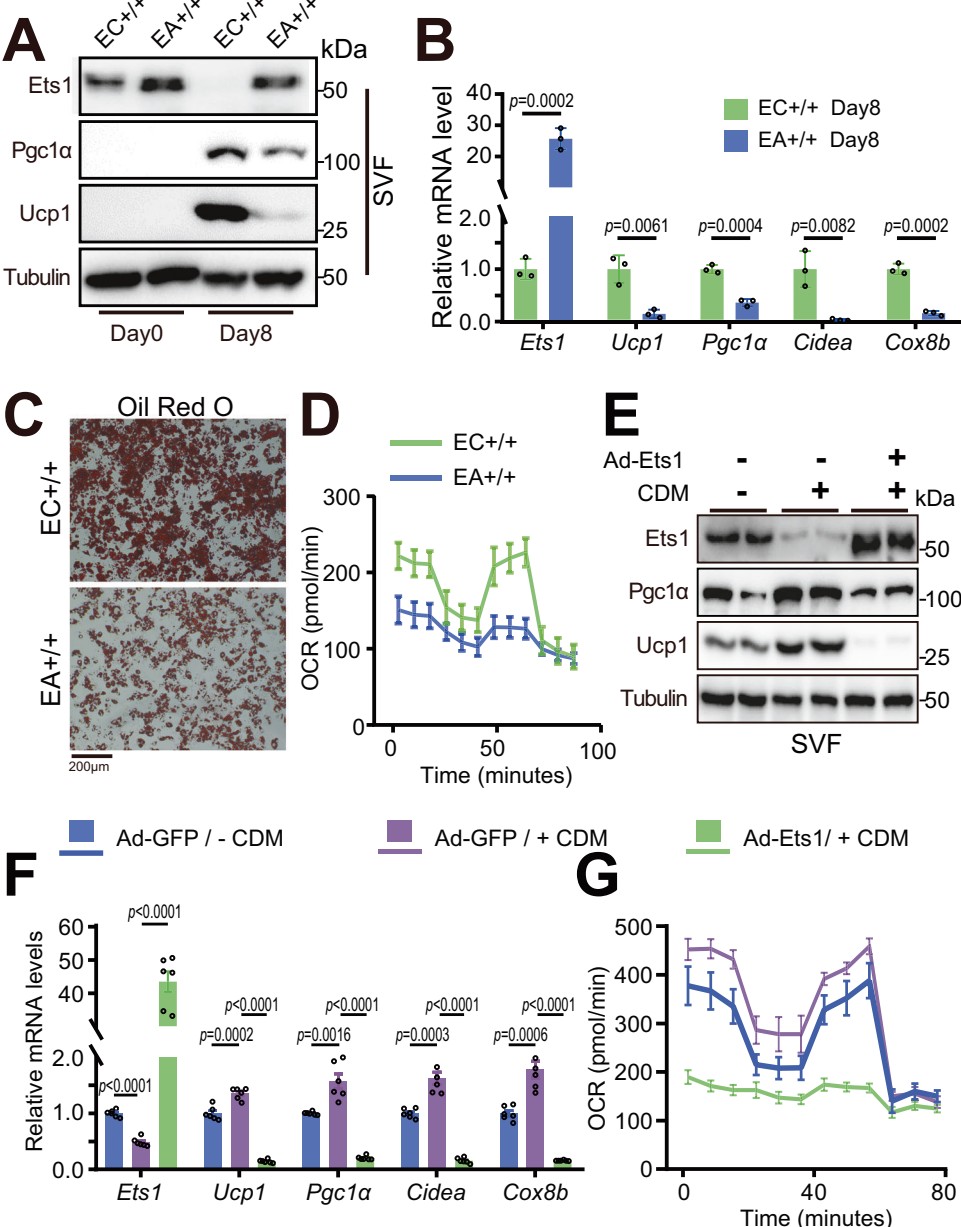

**Fig. 3 | M2 macrophage directly activates adipocytes thermogenesis via blocking Ets1. A–D** SVF cells separated from iWAT of either Ets1[ki/ki] mice (abbreviated as EC+/+) or adipoq-cre × Ets1 [ki/ki] mice (abbreviated as EA+/+) were cultured in vitro and induced to differentiate into beige adipocytes. **A** Western blot assay testing the protein levels of Ets1, Pgc1α, and Ucp1, before (Day 0) or after beige adipogenesis (Day 8) (*n* = 3). **B** qPCR analyzing the thermogenic marker genes of differentiated beige adipocyte (*n* = 3). **C** Oil red O staining of lipid droplets of differentiated beige adipocyte (*n* = 3). Scale bar: 200 μm. **D** Oxygen consumption rate (OCR) of differentiated beige adipocyte (*n* = 8). **E–G** SVFs separated from iWAT

of wild type C57 mice were cultured in vitro. The plate-fully-covered cells were infected with indicate adenovirus for 6 h, then induced differentiation towards beige adipocytes. 5 days later, the cells were treated with 50% M2 CDM for 24 h. **E** Western blot showing the protein levels of Ets1 and thermogenic markers (*n* = 2). **F** Quantitative PCR showing the mRNA levels of Ets1 and thermogenic markers (*n* = 3). **G** OCR measured at Day 6 (*n* = 16). All blot assay was repeated two times independently. Data are means ± SEM. Two-sided Student's *t*-test was used to evaluate statistical significance. Source data are provided as a Source Data file.

expenditure. The EA+/+ mice showed reduced mitochondrial numbers in iWAT after Cl316243 treatment, whereas the numbers in BAT remained unchanged (Fig. 4I). Analysis of the comprehensive metabolic index using a home-cage system in mice stimulated with cl316243 for 7 d revealed no differences in food (SuppFig. 3E) or water intake (SuppFig. 3F). Nevertheless, the carbon dioxide production (VCO2) (Fig. 4J–K), oxygen consumption (VO2) (SuppFig. 3G, H), and heat generation (SuppFig. 3I, J) were lower in the EA+/+ mice than in the control rodents.

## Adipocyte Ets1 knock-out mice are resistant to HFD-induced obesity

The effects of Ets1 on suppressing beige adipogenesis prompted us to investigate whether blocking this protein would modulate global metabolic homeostasis. Adipocyte-specific Ets1 knock-out (EAKO) mice showed reduced Ets1 expression only in adipocyte, while the liver, muscle, or even SVF cell within the adipose tissue were not affected (SuppFig. 4A). When fed a normal chow diet, the EAKO mice showed comparable glucose clearance rates (SuppFig. 3B, C) and

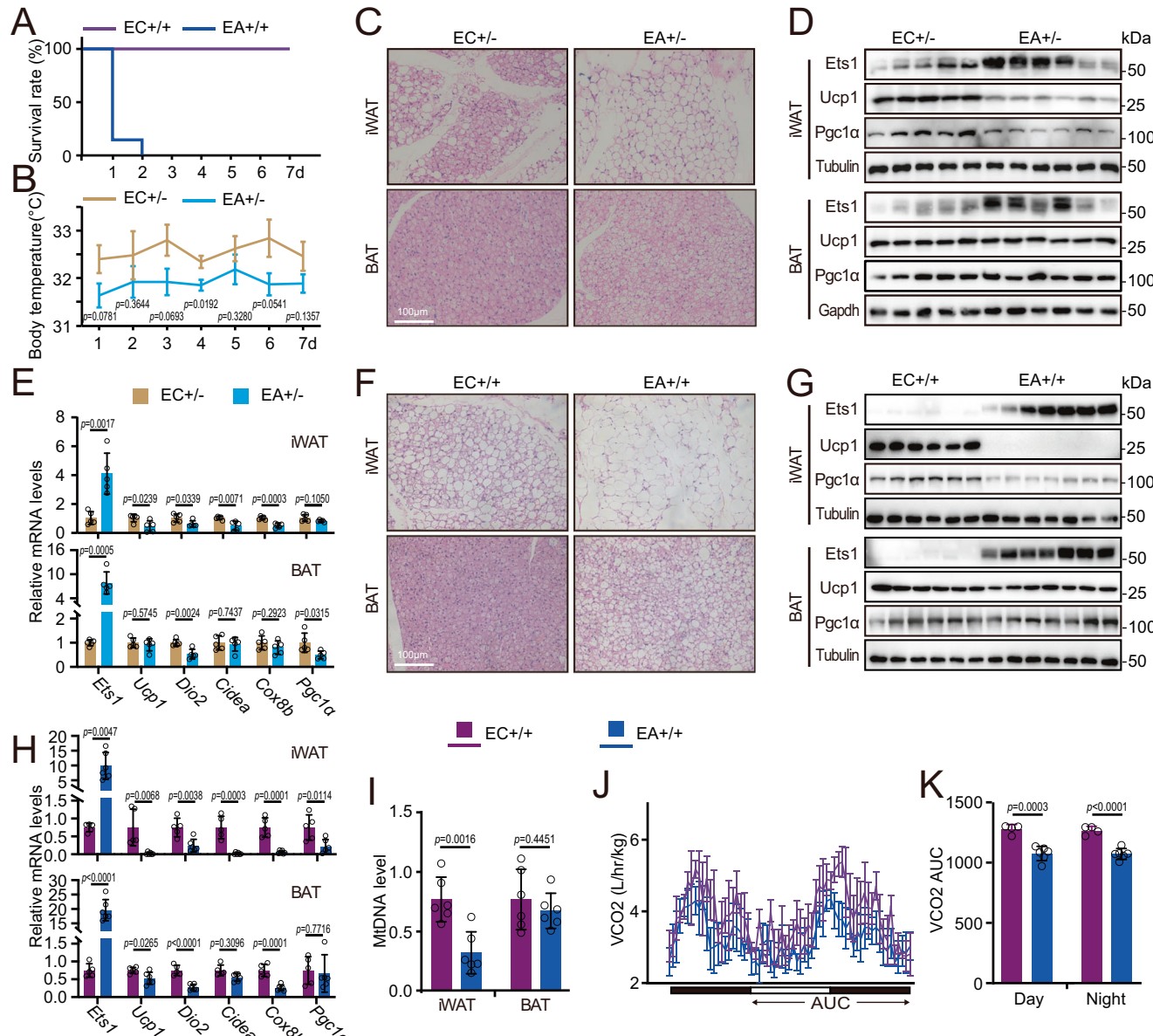

**Fig. 4 | Adipocyte-specific Ets1 knock-in mice are cold intolerance. A** Daily survival rate of EC+/+ (*n* = 6) and EA+/+ (*n* = 7) mice after a 4 °C cold stress. **B–E** Heterozygous Ets1 adipocyte knock-in mice (EA+/-: adipoq-cre × Ets1^ki/-) (*n* = 5) with their control littermates (EC+/-: Ets1^ki/-) (*n* = 6) were treated with a 4 °C cold stress for 7 days. **B** Rectal temperature during cold stress assay. **C** H&E staining of iWAT and brown adipose tissue (BAT) sections. Scale bar: 100 μm. **D** Western blot assay analyzing Ets1, together with the thermogenic markers Ucp1 and Pgc1α. **E** qPCR testing Ets1, and thermogenic fat marker genes. **F–I** Beige adipogenesis was induced in EC+/+ (*n* = 6) and EA+/+ (*n* = 7) mice by daily intraperitoneal injections of 1 mg/kg Cl316243 for 7 days. **F** H&E staining of iWAT and BAT sections. Scale bar: 100 μm. **G** Western blot assay analyzing Ets1, together with the thermogenic

markers Ucp1 and Pgc1α. **H** qPCR testing Ets1, and thermogenic fat marker genes. **I** The mitochondrial DNA (mtDNA) levels of iWAT (left) and BAT (right) in EC+/+ and EA+/+ mice. **J, K** EC+/+ (*n* = 4) and EA+/+ (*n* = 6) were intraperitoneally treated with Cl316243 for 7 days. On day 5, the mice were transferred into the home-cage system, and metabolic parameters from day 7 to day 9 were analyzed. The black and white bars represent night and day, respectively, and the arrow areas were calculated. **J** Carbon dioxide generation. **K** Calculated area under the curve (AUC) of **J**. All blot assay was repeated two times independently. Data are means ± SEM. Two-sided Student's *t*-test was used to evaluate statistical significance. Source data are provided as a Source Data file.

insulin sensitivity (SuppFig. 3D, E), and locomotor activity (Supp-Fig. 3F), to their control littermates. Challenging the EAKO mice with diet-induced obesity led to a reversal of the body weight gain induced by HFD feeding from week 3, and the disparity gradually increased throughout the 14-week experiment (Fig. 5A). After HFD feeding, the EAKO mice showed reduced adipose tissue weight (Fig. 5B) and smaller sizes in all three different fat pads examined (Fig. 5C). Consistently, histological assays suggested that EAKO mice were resistant to HFD feeding-induced white adipocyte hypertrophy and brown adipocyte whitening (Fig. 5D). Compared with the control

littermates, the EAKO mice showed increased expression level of thermogenic marker gene in iWAT (SuppFig. 4G, H) and BAT (SuppFig. 4I, J), consistent with morphological changes.

The beneficial changes in the adipose tissue of EAKO mice corresponded with the whole-body metabolic improvement after HFD feeding. EAKO mice had reduced blood glucose levels in both the fed and fasting status (Fig. 5E), as well as higher glucose tolerance (Fig. 5F, G) and enhanced insulin sensitivity (Fig. 5H, I). In metabolic cage assays, the EAKO mice showed slight increases in food and drink intake (Fig. 5J), together with much higher locomotor activity (Fig. 5K).

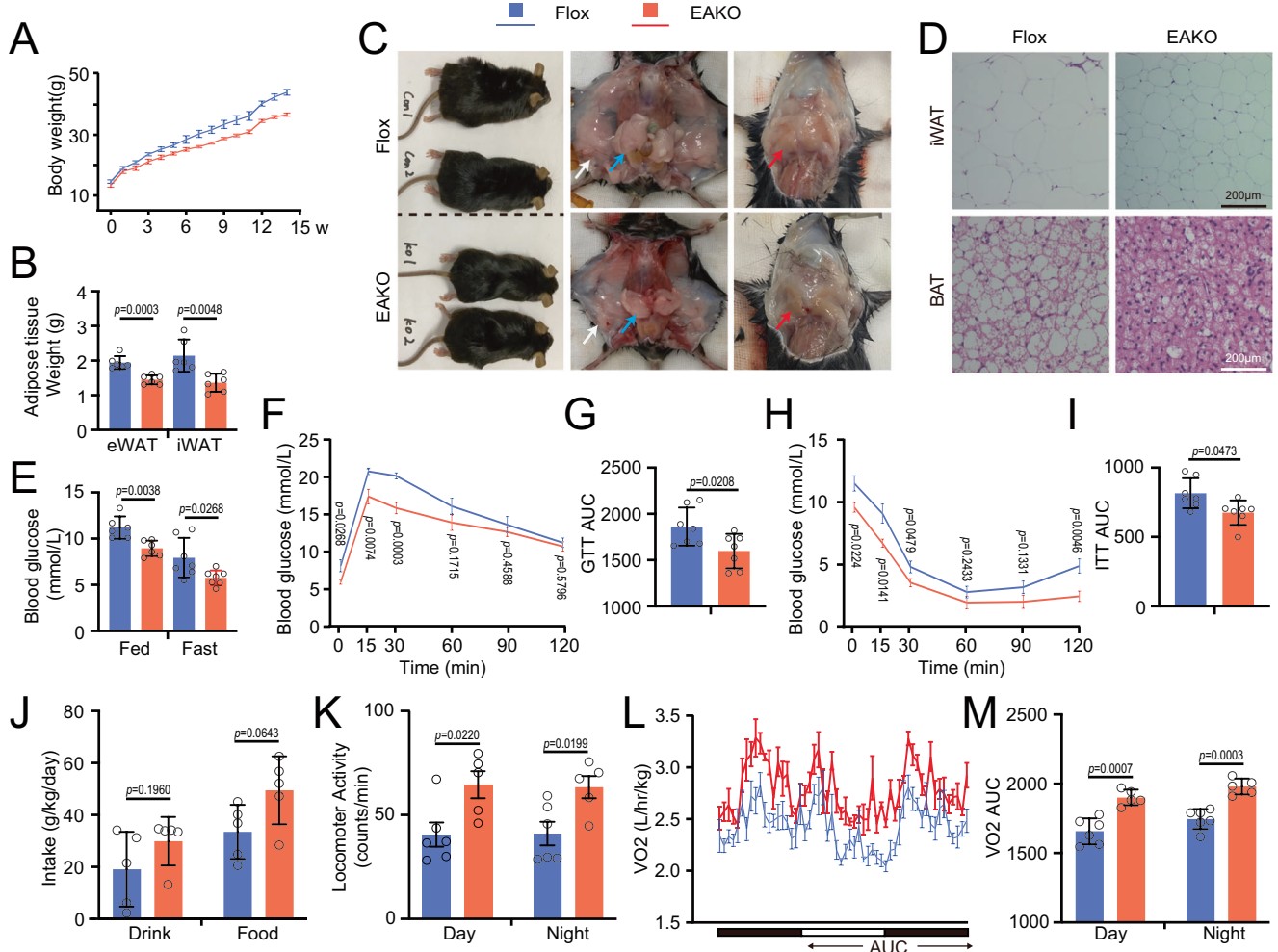

**Fig. 5 | Adipocyte Ets1 knock-out mice are resistant to HFD-induced obesity.**
Ets1 f/f (Flox) and adipocyte-specific knock-out (EAKO) mice at the age of 4-5 weeks were fed a high-fat diet (HFD) for 14 weeks. **A** Body weight curves during HFD feeding ($n = 6$ per group). **B** Adipose tissue weight ($n = 6$ per group). **C** General morphology of the mouse bodies and fat pads. Red arrow, BAT; white arrow, iWAT; blue arrow, eWAT. **D** H&E staining of iWAT and BAT sections. Scale bar: 200 μm. **E** The random and fasting (14 h) blood glucose levels ($n = 6$ for random, $n = 7$ for fasting). **F** Glucose tolerance test (GTT) assay ($n = 7$ per group). **G** Calculated AUC of

**F. H** Insulin tolerance test (ITT) assay ($n = 7$ per group). **I** Calculated AUC of **H**. **J–M** After 14 weeks of HFD feeding, the metabolic parameters of Flox and EAKO mice were analyzed using a home-cage system. The black and white bars represent night and day, respectively, and the arrow areas were calculated. **J** Daily food intake ($n = 5$ per group). **K** Daily locomotor activity ($n = 6$ for Flox, $n = 5$ for EAKO). **L** Oxygen consumption level ($n = 6$ for Flox, $n = 5$ for EAKO). **M** The calculated AUC of **L**. Data are means ± SEM. Two-sided Student's $t$-test was used to evaluate statistical significance. Source data are provided as a Source Data file.

The EAKO mice showed upregulated energy expenditure, as measured by oxygen consumption (Fig. 5L, M), carbon dioxide production (SuppFig. 4K, L), and heat generation (SuppFig. 4M, N).

## Ets1 binds to the promoter region of autophagy-related and mitochondrial genes

We went on to explore the molecular mechanism by profiling the global Ets1 binding site using chromatin immunoprecipitation followed by sequencing (ChIP-seq) analysis. Wild type C57 mice were maintained either at room temperature (RT) or 4 °C (CS) for one week, and the nuclei were separated from mature iWAT adipocytes for library preparation using the Cut & Tag method (Fig. 6A). The sequencing depth and quality were good, and 4056 peaks in the RT samples and 14858 peaks in the CS samples were called (SuppFig. 5A). Peak distribution showed Ets1 enrichment at the ± 1 kb promoter region of its target genes (SuppFig. 5B). Interestingly, cold stress significantly enhanced the overall binding of Ets1 to its target DNA site (Fig. 6B). We therefore focused on the 9905 Ets1 binding peaks that were upregulated after cold treatment (Fig. 6C). KEGG pathway analysis of the genes near the 9905 peaks showed that potential targets

were involved in autophagy and ubiquitin-mediated proteolysis. In addition, many mitochondrial coding genes, subcategorized into Alzheimer disease and Huntington disease, were identified (SuppFig. 5C).

For specific genes, cold stress dramatically increased the overall binding of Ets1 for autophagy-related genes (Atg), such as *Atg2a*, *Atg3*, *Atg5*, *Atg101*, and *Ulk1* (Fig. 6D). Similar effects were also found for genes coding members of all five mitochondrial complexes, namely *Ndufv2* (Complex I), *Sdhb* (Complex II), *Uqcr10* (Complex III), *Cox8a* (Complex IV), *Atp5a1* (Complex V), and for the mitochondrial transcription factor *Tfam* (Fig. 6D). The above binding was further confirmed by conducting Ets1 ChIP-seq using SVF derived beige adipocyte (Fig. 6D). Interestingly, the transcriptional effects of Ets1 on these target genes differed: forced Ets1 expression significantly upregulated the autophagy genes but suppressed the mitochondrial genes (Fig. 6E). By contrast, blocking Ets1 had the reverse effect (Fig. 6F).

## Ets1 transcriptionally suppresses mitochondrial coding genes, as well as activates autophagy-related genes

Ets1 features a unique DNA-binding domain, the ETS domain, which recognizes sequences containing a GGAA/T core motif[39]. We

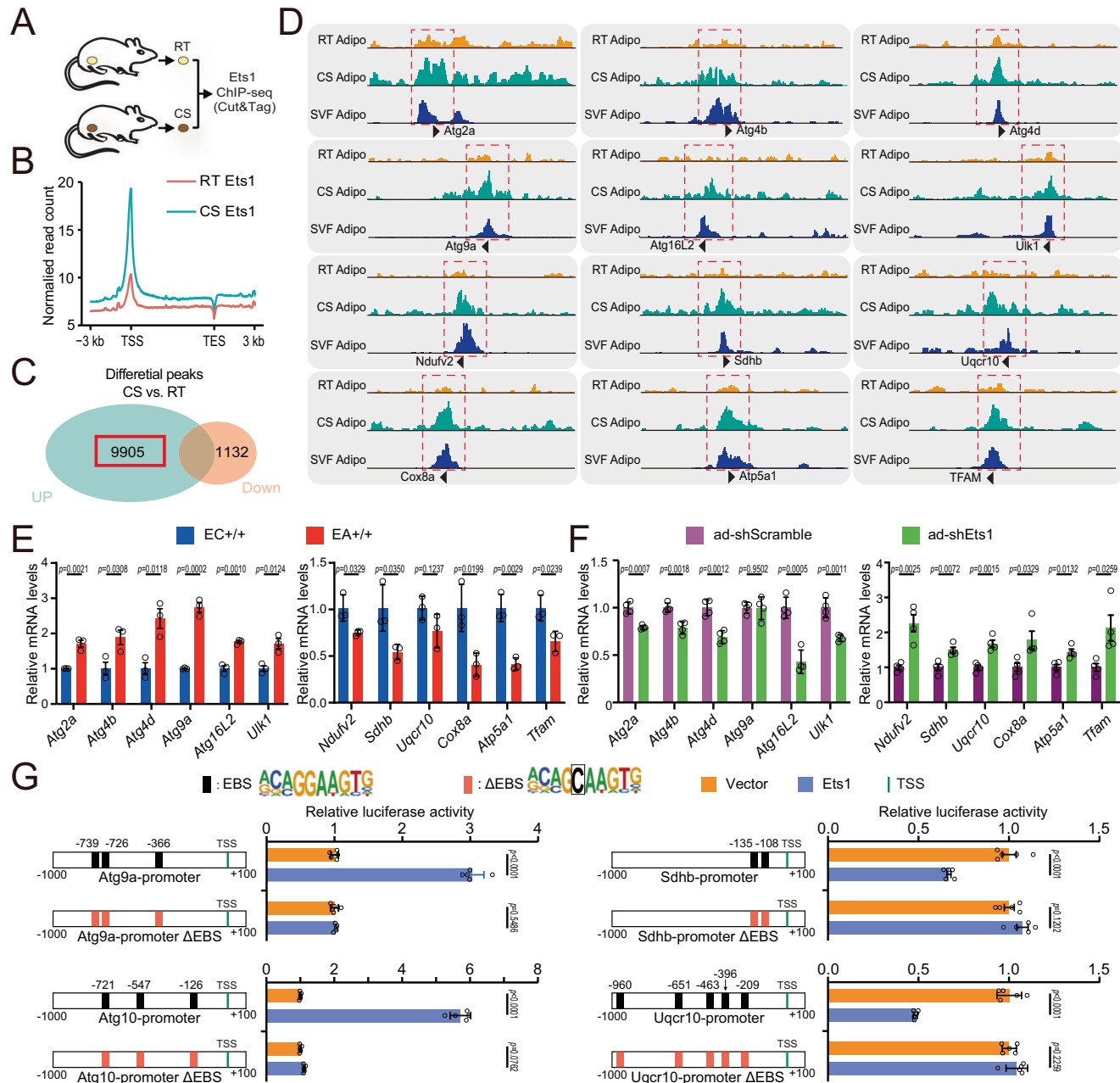

**Fig. 6 | Ets1 transcriptionally suppresses mitochondrial coding genes and activates autophagy-related genes. A** Wild type 8- to 10-week-old C57/BL6j mice were kept either at room temperature (RT) or under cold stress (CS) for 7 days. Mature adipocytes separated from iWAT were used for ChIP-seq (Cut & Tag). **B** Normalized read count (average of reads signals across all genes) across gene body. **C** Venn diagram of peaks. **D** The IGV tool was used to visualize the binding peaks of Ets1 on the promoter region of autophagy-related and mitochondrial genes. Another Ets1 ChIP-seq that using SVF derived beige adipocyte were conducted and together analyzed (SVF adipo). **E** qPCR testing of the mRNA level of autophagy-related (left) and mitochondrial (right) genes, in the iWAT of 8- to 10-week-old EC+/+ or EA+/+ mice (n = 3 per group). **F** qPCR testing of the mRNA levels

of autophagy-related (left) and mitochondrial (right) genes in 3T3L1 cells. Undifferentiated 3T3L1 cells were infected with either ad-Scramble or ad-shEts1 adenovirus; 48 h later, mRNA was collected and analyzed (n = 4 per group). **G** Promoter plasmid construction and the relative luciferase activity. Promoter regions of the mitochondrial clearance genes *Atg9a* and *Atg10* and the biogenesis genes *Sdhb* and *Uqcr10* were cloned, and the core sequence "GGA" in the predicted Ets1 binding sequence (EBS, black box) were point mutated to "GCA" (ΔEBS, red box). Promoter, Ets1 expression plasmid/vectors were co-transfected into 293 T cells, and luciferase activity was tested 24 h later (n = 5 per group). Data are means ± SEM. Two-sided Student's t test was used to evaluate statistical significance. Source data are provided as a Source Data file.

investigated whether Ets1 could change the promoter activity of the target genes through the conserved ETS binding sequence (EBS). Cloning of the wild type promoter regions, from −1000bp to +100 bp, of the *Atg9a*, *Atg10*, *Sdhb*, and *Uqcr10* genes, and conducting luciferase assays revealed that Ets1 overexpression enhanced the transcription of these autophagy genes, but suppressed the activity of the mitochondrial biogenesis genes (Fig. 6G). The transcriptional regulation of Ets1

on these promoter regions was confirmed by point-mutation assays. Mutating the "GGA" sequence located within the predicted EBS to "GCA" completely reversed the regulatory effect of Ets1 (Fig. 6G).

We[31] and others[40,41] have shown that Ets1 binds to epigenetic modulators, such as the histone acetylation "writer" CBP[42], P300[43], and the "eraser" Hdac1[44]. We therefore evaluated the possible epigenetic roles of Ets1 in directing the histone acetylation and regulating the

open chromatin structure in adipocytes. We profiled H3K27ac ChIP-seq and ATAC-seq, using mature adipocytes separated from EAKI mice, EAKO mice, and their corresponding control littermates. Ets1 expression had no effect on the H3K27ac level or accessibility to the promoter regions of the autophagy-related or mitochondrial complex genes (SuppFig. 5D), although it slightly reduced the global ATAC-seq and H3K27ac ChIP-seq read signal (SuppFig. 5E). Taken together, the results showed that Ets1 works as a classic transcription factor, suppressing mitochondrial coding genes, as well as activating autophagy-related genes.

### Ets1 blocks mitochondrial biogenesis and promotes mitochondrial clearance

The precise effects of Ets1 on mitochondrial biogenesis and clearance were analyzed using SVF-derived beige adipocytes. At 0, 2, 4, and 6 days after beige cell adipogenesis, increases were detected in the mRNA levels of complex I-V members, together with the mRNA levels of key mitochondrial transcription factors/cofactors (Fig. 7A). By contrast, forced Ets1 expression completely suppressed these increases in expression of marker genes of mitochondrial biogenesis (Fig. 7A).

In contrast to the inhibitory effect of Ets1 on mitochondrial biogenesis, Ets1 expression promoted mitochondrial clearance. While forced Ets1 expression promoted autophagy (Fig. 7B, C), blocking Ets1 showed an opposite effect in adipocyte (Fig. 7D). Blocking Ets1 also rescued starvation-induced autophagy in 3T3-L1 cells (Fig. 7E). Forced Ets1 expression also activated the PTEN-induced kinase 1 (Pink1)−parkin RBR E3 ubiquitin-protein ligase (Prkn) pathway, a key partner that governs the mitophagy process[45,46], in SVF cells (Fig. 7F). Consistently, immunofluorescence assays revealed far greater co-localization of mitochondria and lysosomes in EA+/+ primary beige adipocytes than in control cells, after mitophagy induction by carbonyl cyanide 3-chlorophenylhydrazone (CCCP) (Fig. 7G). Key factors regulating mitochondria dynamics, such as fusion and fission, were not affected by Ets1 (Fig. 7H).

### M2 macrophage boosts adipocyte mitochondria content and promotes beige adipogenesis via blocking Ets1

ASPC differentiate into beige adipocytes and gain increased mitochondrial content and OXPHOS activity[47]. Our finding that both mitochondrial biogenesis and clearance were regulated by Ets1 prompted further analysis of the effect of Ets1 expression on mitochondrial numbers. Compared with their control littermates, the iWAT from EA+/+ mice contained much lower amounts of the proteins comprising all five mitochondrial complexes (Fig. 8A). Consistently, the EAKO mice showed greater amounts of complex I-V proteins (Fig. 8B).

The mitochondrial inner membrane folds to form cristae, the site at which complex I-V is assembled. The negative regulation of complex I-V by Ets1 was also reflected in the morphological changes in mitochondria, as the EA+/+ mice had fewer mitochondria in their iWAT, and the mitochondria showed disordered structures and reduced amounts of cristae (Fig. 8C). By contrast, the mitochondria from EAKO mice showed well-ordered structure and normal cristae after HFD feeding (Fig. 8C).

The total effect of Ets1 on mitochondrial content was also confirmed using SVF-derived beige adipocytes. At 0, 2, 4, and 6 days after beige cell adipogenesis, forced Ets1 expression significantly reduced the numbers of fluorescently stained mitochondria (SuppFig. 6A), suggesting a blockage of mitochondrial accumulation during beige fat formation.

Since M2 macrophages reportedly increase mitochondrial numbers in beige adipocytes[48], we analyzed whether Ets1 mediated this effect. To mimic in vivo conditions, the adipose tissue macrophages (ATM) were separated via flow cytometry (SuppFig. 6B) and used to collect CDM. While the ATM CDM coculturing significantly raised the mitochondria complex content (Fig. 8D), enhanced the thermogenic marker gene expression (Fig. 8E), activated oxygen consumption rate (Fig. 8F) in the control adipocyte, the Ets1 overexpressing adipocyte showed reversal effect of all the above phenotype (Fig. 8D–F). The promoting effect of M2 macrophage on adipocyte mitochondria content, as well as the mediating role of Ets1 in this process were also confirmed by BMDM CDM coculturing assay (SuppFig. 6C, D).

To further clarify the in vivo function of adipocyte Ets1 in responding to macrophages stimuli, as well as in regulating beige adipogenesis, the M2 macrophage adoption assay was conducted, either on control or on EA+/+ mice. After 4-days cold stress, the adoption group significantly raised the mitochondria content in iWAT (Fig. 8G), thus promoted the cold induced beige adipogenesis (Fig. 8H), up-regulated the body temperature (Fig. 8I), and enhanced the thermogenic gene expression (Fig. 8J) in control littermates, whereas the Ets1 knocking in mice significantly reversed all these effects (Fig. 8G-J).

Taken together, through three independent assays of BMDM CDM co-culture, ATM CDM co-culture, and M2 macrophage adoption, our data discovered a pivotal regulating pathway of M2 macrophage in boosts adipocyte mitochondria content, as well as promoting beige adipogenesis, via blocking Ets1.

## Discussion
Similar to other organs, adipose tissues are subject to immune surveillance. Among the infiltrating immune cells, macrophages dominate and deserve particular attention. Beige adipocytes and macrophages have been recently shown to participate in crosstalk that is important to thermogenic activation. Although adipocytes co-cultured with M2 macrophage showed enhanced beige adipogenesis in vitro[36,48], whether this immune cell alone can effectively activate adipose thermogenesis in vivo is unclear. This is an important knowledge gap not only in clarifying the signals that initiate and maintain beige thermogenesis, but also in understanding how the communication network of adipocytes, SNS, and macrophages is integrated and responses to the ever-changing environmental cues.

Here, we performed chemical sympathectomy in mice, and find that macrophages promote beige adipogenesis independent of the SNS, in agreement with a recent article also showing an SNS-independent effect of macrophages in promoting adipose thermogenesis[49]. The contribution of macrophage-mediated (SNS independent) and SNS-mediated mechanisms in regulating beige adipogenesis remains to be determined, but the present data highlight the effective role of macrophage-adipocyte crosstalk in promoting thermogenesis.

We also identified adipocyte Ets1 as a responder to macrophage stimuli. By analyzing sc-sequencing data, we showed that Ets1 is highly expressed in ASPCs but its expression significantly decreases as the ASPCs differentiate into mature adipocytes. Forced expression of Ets1 in adipocytes can completely reverse the beige-promoting effect of M2 macrophages, thereby highlighting the central role of Ets1-mediated macrophage/adipocyte crosstalk in the modification of thermogenesis. The finding that M2 CDM suppressed the expression of Ets1 in vitro leads us to speculate the upstream signals that modify adipocyte Ets1 expression are independent of SNS activation. Of note, recent studies have shown that, in response to cold stimulus, the skeletal muscle secreted meteorin-like hormone (METRNL) and activated resident eosinophils in fat pad[50], and that these changes further increase the levels of non-inflammatory cytokines IL-4 and IL-13 and promote M2 macrophage recruitment[26]. Although the actual contribution of this process is not yet known, it appears to be a pathway completely independent of the SNS in which cold stress activates eosinophils and M2 macrophages, blocks Ets1 in ASPCs, and promotes beige adipogenesis.

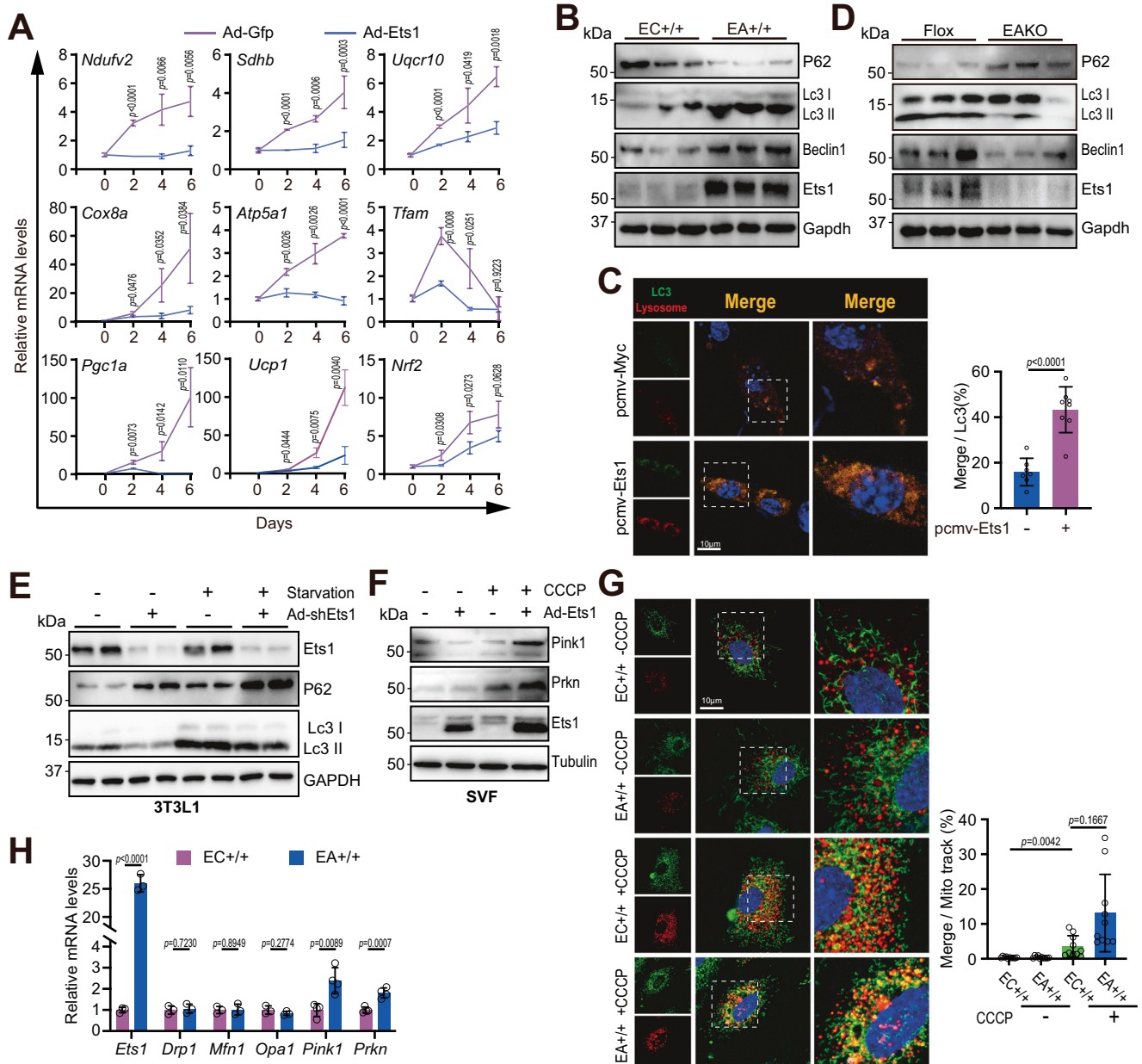

**Fig. 7 | Ets1 blocks mitochondria biogenesis and promotes its clearance. A** qPCR testing of the mRNA levels of Ets1-targeted genes, together with beige marker genes ($n = 3$ per group). SVFs derived from iWAT of C57 mice were infected with corresponding adenovirus. Beige adipogenesis was induced 24 h later (deemed day 0). **B** Western blot analysis of autophagy flux in mature adipocytes derived from iWAT of EA+/+ mice, in comparison with control littermates ($n = 3$ per group). **C** Representative immunofluorescence showing LC3 and lysosomes. 3T3L1 cells were transfected with pcmv-Vector ($n = 7$) or pcmv-Ets1 plasmid ($n = 8$), followed by eGFP-LC3-lysosome adenovirus infection. The ratio of Merge/Lc3 area were calculated (right). Scale bar: 10 μm. **D** Western blot analysis of autophagy flux in mature adipocytes derived from iWAT of EAKO mice, in comparison with control littermates ($n = 3$ per group). **E** Western blot analysis of autophagy in 3T3L1 cells ($n = 2$). Cells were infected with indicated adenovirus; 48 h later, the medium was changed to low fetal bovine serum (FBS) (0.5%, starvation) or fresh normal FBS (10%), and cultured for another 24 h. **F** Western blot testing the protein levels of

Pink1, Prkn, and Ets1 ($n = 3$). SVFs were infected with or without ad-Ets1 adenovirus and analyzed 48 h later. Before sample collection, the cells were treated with 0 or 20 μM carbonyl cyanide 3-chlorophenylhydrazone (CCCP) for 3 h to activate mitophagy. **G** Representative images of Mito-track immunofluorescence and lysosomes in beige adipocytes. SVFs were separated from iWAT of indicated mice and differentiated into beige adipocytes for 48 h. Photographs were taken after CCCP stimulation for another 3 h. Nuclei were stained with Hoechst. The corresponding Merge/Mito-track rates are shown (right). Each group included three biological replicates, and a total of 10 photographs were taken for statistical analysis. Scale bar: 10 μm. **H** qPCR testing of the mRNA levels of mitochondrial dynamic genes in mature adipocytes derived from iWAT of EC+/+ and EA+/+ mice. $N = 3$ per group for *Ets1, Drp1, Mfn1, Opa1*, $n = 4$ per group for *Pink1* and *Prkn*. All blot assay was repeated two times independently. Data are means ± SEM. Two-sided Student's *t*-test was used to evaluate statistical significance. Source data are provided as a Source Data file.

Our study also raises an interesting point, in that Ets1 can act as an integrator that promotes the bidirectional regulation of mitochondrial generation and degradation. Beige adipocytes respond to cold exposure and retain mitochondria, whereas they acquire a white adipocyte-like phenotype and lose mitochondria upon withdrawal of the cold

stimulus. Therefore, beige adipocytes are an ideal model for studying the mechanism of mitochondrial biogenesis and clearance. Application of cold or a β3-AR agonist[51] promotes a dramatic increase in the biogenesis of mitochondria by transcriptional modulation[52–55]. Conversely, genetic deletion assays have suggested that the autophagy

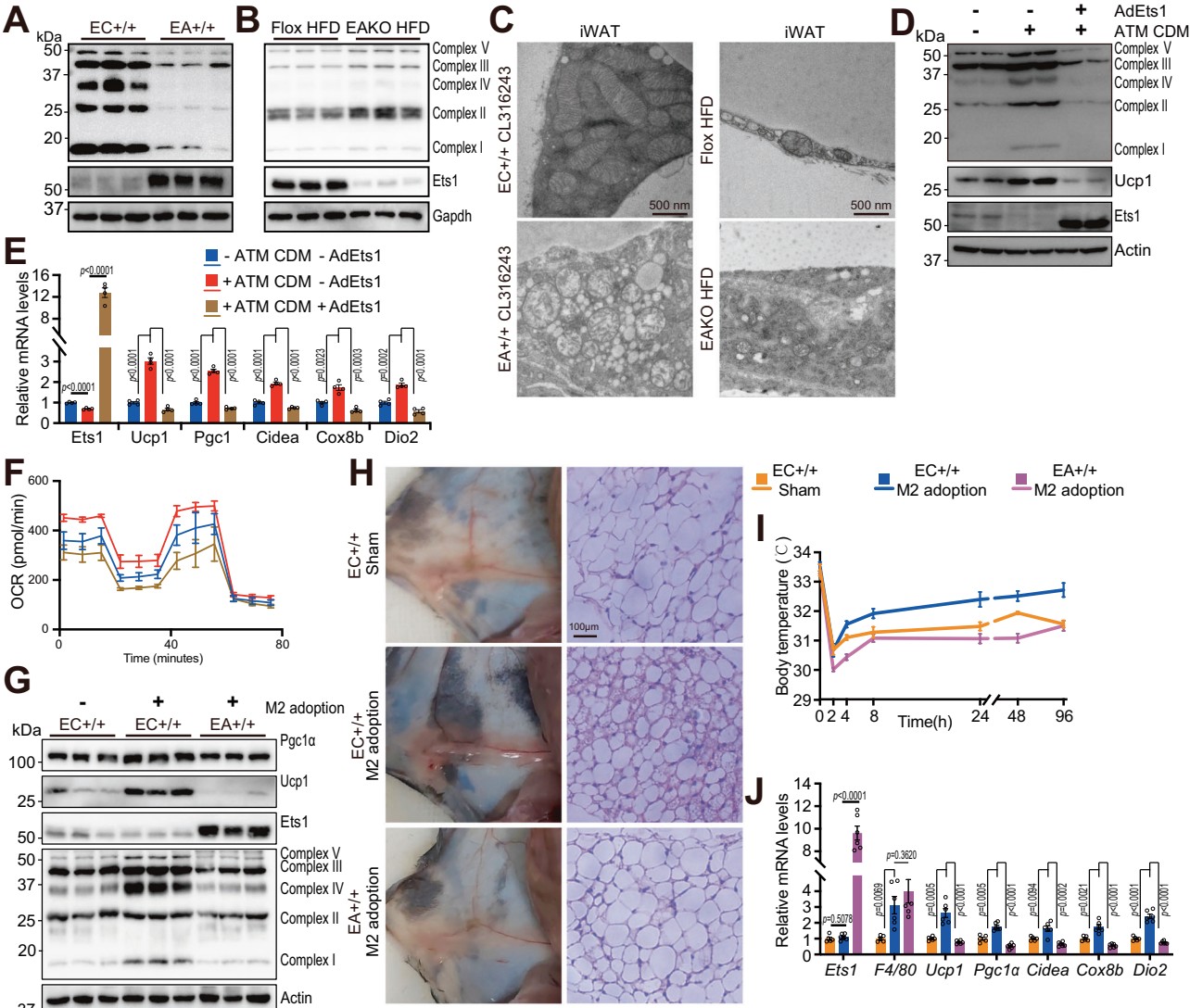

**Fig. 8 | M2 macrophage boosts adipocyte mitochondria content via blocking Ets1.** Western blot analysis of mitochondrial oxidative respiratory chain (OXPHOS) in mature adipocytes derived from in iWAT of Ets1 adipocyte knock-in (**A**) and knock-out mice (EAKO) (**B**), in comparison with corresponding control littermates (*n* = 3 per group). **C** Transmission electron microscopy showing the structure of mitochondria. Left, iWAT samples were separated from EC+/+ and EA+/+ mice, after 7 days of Cl316243 treatment; Right, iWAT samples separated from Flox and EAKO mice, after 14 weeks of HFD feeding. *N* = 3 per group. Scale bar: 500 nm. **D**–**F** SVFs separated from iWAT of wild type C57 mice were cultured in vitro. The plate-fully-covered cells were infected with indicate adenovirus for 6 h, then induced differentiation towards beige adipocytes. 5 days later, the cells were treated with 50% ATM CDM for 24 h. **D** Western blot analysis of the level of OXPHOS, Ets1, and Ucp1 (*n* = 2). **E** Quantitative PCR showing the mRNA levels of Ets1

and thermogenic markers (*n* = 4). **F** OCR measured at Day 6 (*n* = 6). **G**–**J** M2 BMDM were adopted in both side of iWAT, either in EC+/+ or EA+/+ mice. After 6OHDA induced SNS ablation, all mice were then treated with a 4-day cold stress. *N* = 5 for sham group, *n* = 6 for adoption group. **G** Western blot analysis of the level of OXPHOS, Ets1, and Ucp1. **H** General morphology of iWAT (left) and H&E staining of iWAT (right). Scale bar: 100 μm. **I** Rectal temperature measured during old stress. EC+/+ M2 adoption VS. EC+/+ Sham, *p* = 0.6554/0.9632/0.0138/0.0298/0.0159/ 0.0171/0.0030 for 0/2/4/8/24/48/96 h; EA+/+ M2 adoption VS. EC+/+ M2 adoption, *p* = 0.7173/0.0167/0.0001/0.0028/0.0133/0.0001/0.0023 for 0/2/4/8/24/48/96 h; **J** qPCR testing Ets1, macrophage marker F4/80, and thermogenic fat marker genes. All blot assay was repeated two times independently. Data are means ± SEM. Two-sided Student's *t*-test was used to evaluate statistical significance. Source data are provided as a Source Data file.

machinery components, such as Atg5[51,56], Atg7[56–59], and Atg12[51], negatively regulate the beiging process. Although some studies have provided hints that mitophagy and mitochondrial biogenesis are coordinately controlled[60–62], before our work, how do beige adipocytes integrate stimuli and reconcile these two opposing process remains unclear.

Here, we identified Ets1 as a mitochondrial repressor that promotes mitochondrial degradation while also blocking mitochondrial biogenesis. This finding is noteworthy because M2 macrophages reportedly not only boost mitochondria numbers in adipocytes[48], but they also govern the quality control of thermogenic fat metabolism by removing oxidatively damaged mitochondria[63,64]. However, how beige

adipocytes respond to M2 macrophages and govern the quantity and quality of mitochondria is not known. Our ChIP-seq data revealed that Ets1 directly binds to the promoter region of both mitochondrial biogenesis-related and degradation-related genes, with the effect of transcriptionally reducing the mitochondrial numbers. We therefore hypothesize that the M2 macrophage-responsible suppression of Ets1 trigger mitochondria accumulation and promote the adipose beiging process. The reciprocal results we found with Ets1 adipocyte knock-in and knock-out mice confirmed this assumption: Ets1 reduced the levels of oxidative complex proteins, led to morphological abnormalities in the mitochondria, and abolished the mitochondria promoting effect of M2 macrophage. Surprisingly, the suppressing effect of Ets1 on

mitochondria was completely dependent on transcriptional regulation, with no apparent role for epigenetic modification on H3K27ac or chromosome accessibility, although we and others have reported that Ets1 interacts with histone acetylation regulators[30,31,34,40,41]. Although this may seem intuitive, our work demonstrates a unique mechanism by which adipocytes respond to M2 macrophage stimuli and increase mitochondria numbers by blocking Ets1.

Beige adipose accumulates mitochondria, increases energy expenditure and has a protective effect against obesity and consequent metabolic disorders[65]. In this study, we showed that the heterozygous/homozygous Ets1 knock-in mice showed a dose-dependent suppression of beige adipogenesis. Conversely, the Ets1 adipocyte knock-out mice showed an enhanced energy expenditure, gained less weight, and were resistant to HFD induced insulin resistance. Interestingly, locomotor activity was significantly increased in the EAKO mice after HFD feeding; whereas in normal chow diet feeding condition, the locomotor activity was similar between EAKO and Flox mice. This indicates that blocking Ets1 in adipocytes do not directly affect physical activity, at least in the healthy state. Therefore, it seems that the increased motor ability of EAKO mice may be secondary to the effect of weight loss, rather than a direct impact on the muscle. Another interesting point to note is that, compared with its effects in beige adipocytes, Ets1 showed much less effect on Ucp1 expression and thermogenic marker gene activation in BAT. This inconsistency could reflect the diverse origin of these two fat cells: the brown adipocyte shares a progenitor cell (Myf5$^+$) with skeletal muscle, rather than with white adipocytes[66,67].

Despite the limitations discussed above, we believe our data support a paradigm in which M2 macrophages communicate with adipocytes and promote thermogenesis by blocking Ets1, an integral governor of mitochondrial generation and degradation.

# Methods

## Animal model
For thermogenic fat activation, mice were intraperitoneally injected cl316243 (1 mg/kg, sigma, C5976) for seven consecutive days at the same time of a day (~10:00). For cold stress, age matched male littermates were subjected to cold stimulation for 4 or 7 days in a well-ventilated refrigerator. EAKO mice of the same age and sex background, together with their wild-type counterparts (Flox) were provided with either normal chow diet or HFD (D12492, 60% kCal from fat; Research Diets) for 14 weeks. All the mice were kept in a SPF level facility at Nanjing Medical University, and were provided with adequate food and water, as well as normal light, temperature and humidity (12-h light/dark cycle, 60-70% humidity). Before tissue collection, mice were assigned to euthanasia via carbon dioxide inhalation, with a fill rate of 50% of the chamber volume per minute. All animal studies were conducted under the guidance of the Institutional Animal Care and Use Committee of Nanjing Medical University (permit number IACUC-1911030).

## Adipocytes specific knock out/in mice generation
Ets1 knockin was constructed by Cyagen Biosciences (Suzhou, China) with C57BL/6J background. Briefly, the gRNA to Hipp11 locus, the donor vector containing "CAG-loxp-Stop-loxp-mouse Ets1 CDS (without stop codon) −2A-EGFP-polyA" cassette, and Cas9 mRNA were co-injected into fertilized mouse eggs to generate targeted conditional knockin offspring. After confirming correctly targeted ES clones, selecting some clones for blastocyst microinjection, followed by chimera production. Founders were confirmed as germline-transmitted via crossbreeding with Flp-deleter.

Ets1 Knockout mice were generated as mentioned before[31]. The obtained Ets1 knock-in /out mice were propagated with adipoq-cre mice (#028020, Jackson Lab) to obtain the adipocytes specific knock-in and knock-out mice. Homozygous Ets1 adipocytes knock-in mice

(Ets1$^{ki/ki}$ * adipoq-cre, EA+/+) were compared with homozygous control littermates (Ets1$^{ki/ki}$, EC+/+), while heterozygous Ets1 adipocytes knock-in mice (Ets1$^{ki/-}$ * adipoq-cre, EA+/-) were compared with heterozygous control ones (Ets1$^{ki/-}$, EC+/-). Ets1 adipocytes knock-out mice (Ets1$^{f/f}$ * adipoq-cre, EAKO) were compared with control flox mice (Ets1$^{f/f}$, Flox). C57BL/6J mice were purchased from Animal Core Facility of Nanjing Medical University, *db/db* mice were purchased from the Model Animal Research Center of Nanjing University (MARC, Nanjing, China).

The primer sequence of genotype PCR is shown in the Supplementary Table 1.

## Metabolic Cage
The overall oxygen consumption (VO$_2$) and carbon dioxide output (VCO$_2$), Heat generation, daily food and drink intake, were determined by the automated home-cage system (TSE; PhenoMaster). The parameters reflecting the global metabolic capacity of the mice were recorded every 40 minutes on average and tested continuously for over 36 h. VO$_2$, VCO$_2$, heat production and food intake were normalized to the body weight of the starting time point. The area under the curve (AUC) of each mouse was calculated separately and the average value of each group was then compared.

## Body temperature
The rectal temperature of the mice was measured with a portable digital thermometer (SH610S, Zhongshenghong-Technology, Beijing, China), and the metal probe was inserted at the same depth each time, rectal temperature was recorded after waiting about 3−5 seconds for the readings to stabilize. Mice were measured at a fixed time each day.

## Glucose tolerance test and insulin tolerance test
The glucose tolerance test (GTT) was performed by intraperitoneal injection of glucose (2 g/kg for NCD fed mice; 1 g/kg for HFD fed mice) in mice fasted over 14 h. The insulin tolerance experiment (ITT) was performed by intraperitoneal injection of insulin (0.75 U/kg for NCD fed mice; 1 U/kg for HFD fed mice) to mice fasted for 4 h. The blood from tail vein of the mice were used to test glucose levels at 0 / 15 / 30 / 60/90/120 min after glucose or insulin injection, via a glucometer (Accu-Chek, Roche).

## Survival rate
To calculate the survival rate of mice, the number of mice that survived each day was recorded. The survival of the mouse was recorded as 1 and the death was recorded as 0. The survival of mice was graphed using the survival curve part of GraphPad Prism 8 (GraphPad Software).

## Sympathectomy and macrophage clearance
For sympathetic denervation, mice at the age of 8-10 weeks old received intraperitoneal injection of 100 mg/kg 6-hydroxydopamine (6-OHDA) dissolved in 0.9% NaCl plus 10$^{-7}$ M ascorbic acid on day −5, −3, and −1. At day 0, the mice were anesthetized with isoflurane and shaved, then each side of the subcutaneous fat was injected 4 times (2 for each side) with 50 μL clodronate liposomes or control liposomes into iWAT, via subcutaneous injection without surgical incision, while the control mice received equal amount of negative liposomes. The injected mice were then cold stressed at 10 °C for 6 h and then 4 °C for 4 days. At day 2, the clodronate liposomes / negative liposomes injection were repeated once. At day 4, the fad pads were collected and snap-frozen in liquid nitrogen for measurement of NE content and F4/80 expression, to confirm the efficiency of denervation and macrophage clearance.

## Histology
Fat pads were isolated from mice, fixed with 4% paraformaldehyde overnight, paraffin-embedded sectioned at the thick of 5μm, rehydrated and stained with hematoxylin and eosin (HE). All sections were photographed by Olympus microscope.

## Transmission Electron Microscope

Adipose tissues were removed from mice and fixed with 2.5% glutaraldehyde dissolved in 0.1 mol/L sodium cacodylate buffer. The tissues were then fixed in 1% $OsO_4$, 1.5% $K_4Fe(CN)_6$, and 0.1 mol /L sodium cacodylate for 1 h. The tissue blocks were stained, dehydrated, and embedded for ultrathin sections (60-80 nm), followed by transmission electron microscopy imaging. All samples were captured by transmission electron microscope (FEI Tecnai G2 Spirit Bio TWIN).

## Primary SVF separation and differentiation

SVF cells were obtained from adipose tissue in the inguinal position of male C57BL/6 J mice at 3-4 weeks old. Type I collagenase (2 mg/ml, Sigma, SCR103) was preconfigured using DMEM basic medium (Gibco, C11995500BT) containing 2% volume of FBS (Gibco), 100 U/ml penicillin, 100 µg/ml streptomycin, and stored in the dark after removal of bacteria by filtration using a 0.22 µm filter. The volume ratio of collagenase to adipose tissue was maintained at 2:1. Then the adipose pad was cut into pieces to 1 mm³ and digested in 37 °C water bath for 1 h, during which the tube was reversed and mixed every five minutes. The digested adipocytes were filtered (70 µm), then centrifuged at 180 g for 10 min. After re-suspension with PBS, centrifuge at the same speed and time. DMEM/F-12 (Meilunbio, MA0214) containing 10% FBS (Gibco) and 1% penicillin-streptomycin was used to culture the primary SVF, and then the cells were implanted into the pore plate of appropriate size. The medium was changed 24 h later, and then every 48 h until the cells grew to contact inhibition state.

The plate-fully-covered cells then subjected to differentiation. As for beige adipogenesis, culture medium was added insulin (0.5 µg/ml, sigma, I3536), Dexamethasone (5 µM, sigma, D4902), 3,3′,5-Triiodo-L-thyronine (T3, 1 nM, sigma, T2977), Rosiglitazone (1 µM, sigma, R2408), Isobutyl methylxanthine (0.5 mM, sigma, I5879). The above differentiation medium was changed 48 h later with medium only containing insulin, T3 and rosiglitazone, and renewed every 48 h. Lipid droplet formation was observed at day 7-8, indicating successful induction of beige adipogenesis.

As for white adipogenesis, insulin (1 µg/ml), Dexamethasone (1 µM), Isobutyl methylxanthine (0.5 mM) were added into the same induction medium. The medium was changed 48 h later, only containing insulin to maintain the differentiation of adipocytes.

Cells were incubated at 37 °C in a suitable atmosphere containing 5% $CO_2$.

## M2 macrophage conditional medium

Bone marrow cells were isolated from 8-week-old male wild type C57BL/6j mice, as previously described[68]. The cells were then cultured for 7 days to retain only the bone marrow-derived macrophages (BMDMs), using the medium containing 10% FBS, 10 ng/ml M-CSF, and antibiotics. The BMDMs were treated with IL4 (10 ng/mL) for 24 h, to achieve M2 polarization. The medium were then changed to DMEM with only 10% FBS, and antibiotics, and 24 h later, the generated conditional medium were then collected and used to treat adipocytes.

## M2 macrophage adoption

For M2 macrophage adoption, male mice at 8-week-old were used. The BMDMs were treated with IL4 (10 ng/mL) for 24 h, to achieve M2 polarization. BMDMs were then digested and cell counted after being suspended with PBS. 1 ml sterile medical syringes were used to inject M2 macrophages into iWAT on each side of mice at day 1, 3 and 5 respectively, ensuring that the amount of each injection was $10^6$. The control group was injected with the same volume of PBS. The mice were then given 6-OHDA injections at days 5, 7, and 9 to operate sympathectomy, as previously described. At day 10, the injected mice were cold-stressed at 10 °C for 6 h and then 4 °C for 4 days, after which samples were collected. All mice were provided with adequate food and water during feeding.

## Adipose tissue macrophage conditional medium

Adipose tissue macrophages were obtained from adipose tissues in the inguinal and epididymal position of male C57BL/6 J mice at 10-week-old. Digestion of adipose tissues with Type I collagenase to obtain SVF cells (including macrophages). Cells were incubated with RBC lysis buffer for 5 minutes, adding PBS in order to neutralize. The cells were then centrifuged at 500 g for 5 minutes and resuspended in PBS. Fcγ receptor blocker (Biolegend, 101319) was used to incubate cells for 20 minutes before staining, then washed cells with PBS and stained with Fixable Viability Dye (Biolegend, 423101). Repeat washing and resuspended cells in PBS, stained with indicated fluorescent-conjugated antibodies for 30 minutes. The adipose tissue macrophages were then sorted using flow cytometry, related antibodies included PE-anti-F4/80(Biolegend, 123110) and APC-anti-CD11b (Biolegend, 101212). The adipose tissue macrophages were then cultured for 24 h, using Iscove's Modified Dulbecco Medium (IMEM) with 10% FBS and antibiotics, adding IL4 (10 ng/mL) to maintaining M2 polarization. The generated conditional medium was then collected and used to treat adipocytes.

## 3T3-L1 and Cos7 cell culture and treatment

3T3-L1 (#CL-173) and Cos7 (#CL-1651) were acquired from ATCC (Manassas, VA).

The mouse embryonic fibroblast cell line 3T3-L1 was cultured with high-glucose DMEM (Gibico) containing 10% CBS (Gibico) and 1% penicillin-streptomycin. The renal fibroblast cell line from African green monkey Cos7 was cultured with high-glucose DMEM (Gibico) containing 10% FBS (Gibico) and 1% penicillin-streptomycin. All cells were cultured at 5% $CO_2$, at 37 °C.

In adenovirus infection, the virus was mixed with DMEM in proportion and incubated for 4-6 h, the medium were changed and the expression was allowed for 48 h. As for plasmid transfection, cells were culture for 24 h after transfection using Lipofectamine 2000 (Invitrogen, 11668500), according to the instruction.

## Immunofluorescence

For immunofluorescence live cell staining, the corresponding live cell dye was mixed into the complete medium in proportion. After 15 min of incubation, the cells were washed with PBS for two times and then replaced with new complete medium. Images were captured and analyzed by a confocal laser scanning microscope (Olympus FV1200). The viable cell dyes used and their corresponding working concentrations were as follows: Mito-tracker Green (200 nM in complete medium, Life technologies, M7514); Mito-tracker Red (200 nM in complete medium, Life technologies, M7512); Lyso-tracker Red (400 nM in complete medium, Life technologies, L7528); Hoechst (0.5 mg/mL in complete medium, Sigma, 14533).

For immunofluorescence staining of paraffin sections, iWAT were washed with ice cold PBS and fixed in 4% PFA at 4 °C for 12 h. After paraffin embedding, the samples were cut into 10-µm slices and dewaxed, rehydrated and permeated. After being restored to room temperature, the sections of iWAT were first soaked in PBS for 10 min and then blocked by 3% w/v BSA for 30 min. The primary antibody against tyrosine hydroxylase (1:500 dilution, 1% w/v BSA, Proteintech, 25859-1-AP) and Alexa Fluor 488 conjugated secondary antibodies (1:500 dilution, 1% w/v BSA, Thermo Fisher Scientific, A-11001) were uniformly added to the sections, incubated at room temperature for 1 h. Images were captured and analyzed by a confocal laser scanning microscope (Olympus FV1200).

## Cellular oxygen consumption rate

Cellular oxygen consumption was measured by Seahorse Bioscience XF96. Adipocytes were plated and differentiated in XF96 cell culture microplates. Prior to analysis, adipocyte culture medium was changed to respiration medium consisting of DMEM lacking $NaHCO_3$ (Sigma), NaCl (1.85 g/L), phenol red (3 mg/L), 2% fatty-acid-free BSA, and

sodium pyruvate (1 mM), pH 7.4. Under these conditions, basal respiration is OCR in the presence of the substrate (1 mM sodium pyruvate) only, respiration uncoupled from ATP synthesis was determined following addition of oligomycin (1 μM). Maximal respiration was determined following addition of FCCP (1.5 μM). Antimycin A and Rotenone (0.5 μM each) were used to abolish mitochondrial respiration. The data was subsequently exported using Seahorse Wave software.

## Cellular oil red O staining

Before oil red O staining, the cells were fixed with 4% paraformaldehyde for 15 minutes. Then, the saturated oil red O dye was mixed with ddH$_2$O at a ratio of 3:2, and the dye was filtered with a 0.45 μm filter. The cells were stained for 15 minutes, and then cleaned with PBS for 3 times. All cells were photographed by Olympus microscope.

## ChIP-seq

ChIP-seq was perfomed using CUT&Tag strategy[69]. Briefly, native nuclei were purified from fresh mature adipocytes as previously described[70] and were washed twice gently with wash buffer (20 mM HEPES pH 7.5; 150 mM NaCl; 0.5 mM Spermidine; 1× Protease inhibitor cocktail). The following tagmentation, library construction and sequencing were performed by Shanghai Jiayin Biotechnology Ltd. In brief, 1×10^5 nuclei bound to Concanavalin A-coated beads were incubated with indicated primary antibodies (Ets1, Active Motif, #39581, 1:50; H3K27ac, Active Motif, #91193, 1:50; normal rabbit IgG, Merck Millipore #12-370; normal mouse IgG, Merck Millipore #12-371) and second antibodies (anti-rabbit IgG antibody, goat monoclonal: Millipore AP132; anti-mouse IgG antibody, goat polyclonal: Millipore AP124). Transposase pA-Tn5 was used to induce DNA tagmentation, followed by Proteinase K digestion and PCI extraction. The ethanol-precipitated DNA was immediately subjected to PCR amplification by Universal and barcoded i7 primer using NEBNext HiFi 2x PCR Master mix. The amplified library was purified without size selection. ChIP-seq libraries were prepared from 3-5 ng DNA using NEBNext® Ultra™ II DNA Library Prep Kit (NEB, E7645S), according to the manufacturer's instructions. Post-amplification libraries were size selected at 250–450 bp in length using AMPure XP beads ((Beckman Counter)) and were quantified using the Library Quantification Kit for Illumina (Kapa Biosystems, KK4603). Libraries were pooled to a final concentration of 10 nM and sequenced paired-end using the Illumina NovaSeq 6000.

## ATAC-seq

Nuclei extracts were prepared as described in ChIP-seq, and the followed ATAC-seq was performed according to ATAC-seq protocol by Shanghai Jiayin Biotechnology Ltd. In brief, the Nextera DNA Library Preparation Kit (Illumina) was used to perform the transposition according to the manufacturer's manual. 50,000 nuclei were pelleted and resuspended with transposase, for 30 minutes at 37 °C. The transposed DNA fragments were purified immediately after with a MinElute PCR Purification Kit (Qiagen). After Samples were PCR-amplified using 1X NEBNext High-Fidelity PCR Master Mix (New England Biolabs, MA). Subsequent libraries were purified with the MinElute PCR Purification Kit (Qiagen) and subjected to sequencing on Illumina Novaseq 6000.

## Data processing of ChIP-seq and ATAC-seq

We followed the bioinformatic pipeline described in to process the sequencing reads. Before pair-end reads mapping, clean reads were obtained from the raw reads by removing the adaptor sequences, by trimmomatic software[71]. The clean reads were then aligned to reference mm10 genome sequences using BWA program[72]. We used macs2[73] to call peaks and selected peaks with cutoff q value < 0.05. The bam file generated by the unique mapped reads as an input file, using deeptools software[74] for bigwig generation. The deeptools tool is used for plot the reads distributions (from bigwig) across peaks, followed by annotating the peaks by the function of annotatePeak of ChIPseeker[75]. The HOMER's[76] tool was used for Motif analysis.

We analysis differential peak through 3 steps. First merge the peak files of each sample using the bedtools software[77]. Second, the counts of the reads over the bed was determined for each sample using bedtools multicov. Finally, differential accessible peak was assessed using DEGseq[78]. Region were called differentially accessible if the absolute value of the log2 fold change was 1 at a P value < 0.05. IGV tool were used for visualization.

Network was built based on the relationship between GO Terms to provide user friendly data navigation and visualization. We selected the significant Terms (P value < 0.05) in Enrichment Analysis based on the up, down and all differentially expressed genes to construct the network to summarize the function affected in the experiment.

## Data processing of the published sc-seq and RNA-seq dataset

The scRNA-seq expression matrix (GSE176171)[8] was processed using the R package Seurat[79]. UMAP reduction was used for cluster visualization, and the R package ggplot2 was used to visualize gene expression. Using the Slingshot algorithm with default parameters, a pseudo-time trajectory analysis was performed to arrange cells into a developmental trajectory, which was segmented with different branches to imitate cell evolution or differentiation.

As for RNA-seq, the expression of screened active DBDs were profiled using a published dataset (GSE228094). The processed RPKM values of all Samples were downloaded and R package ggplots2 was used to generating the heatmap.

## Luciferase assay

Promoter region at −1000bp ~ +100 bp of mitochondria clearance gene *Atg9a*, *Atg10*, and biogenesis gene *Sdhb*, *Uqcr10* were cloned into pGL3-basic plasmid. The core sequence "GGA" in the predicted Ets1 binding site (EBS) was point mutant to "GCA" (ΔEBS), to avoid Ets1 recognition. Wild type promoter or point mutant promoter, Ets1 expression plasmid, and renilla were co-transfected into 293 T cell. The luciferase activity was determined 24 h after transfection, using a Dual-Luciferase reporter assay kit (Promega), according to its instruction. The relative luciferase activity was normalized with renilla activity.

## Western blot

Protein samples were obtained from fresh tissues or cells, both lysed with ice-cold lysis buffer containing 50 mmol/L Tris-HCl (pH 7.4), 150 mmol/L NaCl, 1% NP-40, 1 mmol/L EDTA, 1 mmol/L phenylmethylsulfonyl fluoride, with Complete protease inhibitor (1 tablet/ 10 mL; Roche). Protein content was determined using BCA Protein Assay Kit (Takara, T9300A). Adjust the protein concentration to same level and boiled samples for 10 min. Samples are uniformly processed by SDS-Page gel electrophoresis at a loading amount of 30 μg. The protein was transferred to the methanol pre-activated PVDF membranes (Millipore Corp, USA) and then blocked with 5% skim milk, and followed by incubation with corresponding first and second antibodies. The protein band was visualized using High-sig ECL (Tanon, 180-5001). As for detecting the protein level of mitochondrial complex (OXPHOS), it was only heated to 37 °C.

The detailed information of antibodies are shown in the Supplementary Table 2.

## RNA extraction, Real-time PCR, and RNA sequence

Trizol reagent (Invitrogen) was used to extract total RNA from tissues and cells. 1 μg RNA was reverse transcribed into cDNA by the reverse transcription kit HiScript II Q RT SuperMix (Vazyme, R222-01). After mixing with ChamQ SYBR qPCR Master Mix (Vazyme, Q321-02) in

proportion, quantitative PCR was performed using Light Cycler 480 II Sequence Detection System (Roche, Basel, Switzerland). The RNA sequences of the primers used are available in Supplementary Table 3.

## Statistical analysis

The data presented in this paper were expressed as mean ± SEM. Significant differences are assessed using a two-tailed Student $t$-test or one-way ANOVA for multiple-group comparison. All data analysis was performed using GraphPad Prism 8 (GraphPad Software). P < 0.05 was considered statistically significant.

## Reporting summary

Further information on research design is available in the Nature Portfolio Reporting Summary linked to this article.

## Data availability

The generated raw datasets, together with the analyzed bigwig and narrowpeak files during the current study are available in the GEO repository GSE221335 and GSE234879. In addition, this study also performed analysis using the existing public sequencing data GSE228094 and GSE176171 from the GEO database. The other source data generated in this study are provided in the Source Data file. Any additional information is available upon request to the corresponding author (Kai Li, likai87@njmu.edu.cn) Source data are provided with this paper.

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

## Acknowledgements

We thank Dr. Yangyang Wu (Nanjing Maternal and Child Health Hospital) for guiding SVF separation and in vitro differentiation. We thank Dr. Haoran Sun (Nanjing Medical University) for his material sharing and technical guidance on mitophagy staining. We thank the Animal Facility, the Analysis and Measurement Center of Nanjing Medical University for their help. This work was supported by grants from the National Natural Science Foundation of China (82330027 to X.H.; 82070897 to K.L.; 82171729 to C.W.; 82270849 to P.S.).

## Author contributions

S.W., C.Q., K.L., and X.H. conceived and designed the study; S.W., C.Q., J.N., W.G., J.S., X.Y., Y.C., and Y.S., performed the experiments and generated the data supported by Y.Z., P.S., X.C., and C.W.; S.W. analyzed

the data with help from K.L.; K.L. wrote the manuscript. All authors discussed the results and commented on the manuscript.

## Competing interests

The authors declare no competing interests.

## Ethics

All animal study was done in compliance with and approval from the Institutional Animal Care and Use Committee (IACUC) under the permit number 1911030 of Nanjing Medical University.
