## [Peer Review File · Nature Communications]

M2 macrophages independently promote beige adipogenesis via blocking adipocyte Ets1REVIEWER COMMENTS

Reviewer #1 (Remarks to the Author):

In this study, authors identified that Ets1 is a key mediator of beige adipogenesis stimulated by M2 macrophages. They found that Ets1 is expressed in SVF but not in mature adipocytes. However, when using Ets1 adipocyte specific knockin and knockout, they found that Ets1 affected adipocyte specific mitochondrial density and oxidative function. While data are potentially important and interesting, additional studies are needed in order to conclusively test the mediatory roles of Ets1.

The following are major comments:

1. Since Ets1 is only expressed in SVF, not mature adipocytes (Fig. 2C), why did adipocyte specific knockout and knockin were selected? This question is partially alleviated by the Fig. 2D, where appeared that Ets1 is expressed during early adipogenesis. However, there is still a question about the biological effects of Ets1, whether it is mainly through promoting beige adipogenic commitment and differentiation, or simply through enhancing mitochondrial density and function?
2. 6OHDA was used to induce sympathetic nerve ablation. However, 6OHDA induces broad changes in neurons, not only sympathetic nerves. To be conclusive, additional method such as physical methods may be needed.
3. Similarly, clodronate liposome was used for locally ablate the macrophages, which is insufficient. The study argues that M2 macrophages, not M1 is responsible. Therefore, additional testing in M1 and M2 transition and their effects on Ets1 expression and beige adipogenesis is needed.
4. In figures 4 to 7, Ucp-1 was not analyzed. However, Ucp-1 expression is a critical marker of beiging. Therefore, related analyses should be added.
5. Just for easy to read, in figures, abbreviations, especially unstandardized, should be minimized and treatments/experimental design can be briefly described in the figure legends.
6. I could not find the Method Section and, thus, cannot evaluate experimental procedures.

The following are detailed comments:

Line 60: The reference 14 is a review paper, original study is needed here.

Line 83: Reference is needed after M2 macrophage stimuli.

Line 91: How about the nonspecific effects of 6OHDA?

Line 98: Clodronate liposome ablation removes M2 macrophage specifically? How about M1 macrophage? An additional method which can alter the polarization of macrophages is needed.

Line 191: EA+/+, Ets1ki/ki×Adipoq-cre can be changed to Ets1ki/ki×Adipoq-cre; abbreviated as EA+/+.

Line 245: Changes in beiging need to be analyzed.

Line 251: Why only WT mice? This should be done in Ets1 knockin and knockout mice.

Line 293: It is claimed that Ets1 works as a classic transcription factor suppressing mitochondrial coding genes. If this is the case, it is necessary to analyze additional tissues enriched with mitochondrial, such as muscle, liver, etc.

Figure 1D: Why did F4/80 differ so dramatically due to CL?

Figure 1N: The meaning of horizontal changes needs to be labeled in panel N. Pseudotime started as mature adipocytes and ends in ASPC?

Figure 2C: Panel C showed that the Ets1 is mainly expressed in SVF, not adipocytes. Then why was Adiponectin-Cre was used? Why Ets1 inhibits mitochondrial biogenesis in mature adipocytes if it is not expressed?

Figure 2: Could an additional approach which can inhibit M2 macrophage transition be used to further verify? N=2 is too low.

Figure 4J-M: How about heat production?

Figure 5D: It will be more informative if the length of promoter region can be indicated, including TSS.

Figure 5F: How about UCP-1 and beige related genes?

Figure 5G: Why not UCP-1, the most important gene for beige adipogenesis?

Figure 5H-I: Why mature adipocytes?

Figure 7: This study appears not related to M2 macrophage? A brief description of experimental

design in figure legend can help.

Figure 7F-G: Should also be separated from EA+/+ and EAKO mice?

Reviewer #2 (Remarks to the Author):

The manuscript by Wu et al describes the very interesting finding that M2 macrophages can promote thermogenic beige adipogenesis independently of their effect on the sympathetic nervous system and that this effect is dependent on suppressing the expression of transcription factor Ets1 in adipocytes (particularly in adipose stem and progenitor cells (ASPC)). The data presented are generally convincing and there is a dramatic effect loss of Ets1 in adipocytes resulting in the complete loss of cold tolerance in mice.

The results are overall interesting, but it would be nice to know how M2 macrophages affect Ets1 levels in adipocytes. Does one of the cytokines released by M2 cells, such as IL-6, IL-1 β or TNF- α , affect Ets1 levels?

In Figure 1E, what is the effect of macrophage depletion alone as opposed to macrophage depletion in the context of SNS ablation?

How can you tell macrophage depletion by clodronate is limited to iWAT? It is not clear that the liposomes will only be found in iWAT and not spread systemically.

There is a sentence saying "Consistently, the effects of Ets1 overexpression on BAT thermogenic activation were relatively small" in lines 216-217. However, the histology of the BAT with Ets1 overexpression in figure 3F seems to be significantly different than the control. Perhaps since BAT has more mitochondria to begin with, this tissue can withstand better the effects of Ets1 overexpression? Can the function of the BAT be more directly tested?

Can the authors speculate why EAKO mice show increased locomotor activity?

The sentence in lines 256-257 stating that "cold stress significantly enhanced binding affinity of Ets1" should be revised. The CUT&TAG assay used doesn't assess transcription factor affinity for DNA. Instead, it is assessing overall level of binding of Ets1 to DNA, which can be affected by multiple parameters including increased concentrations of Ets1 in the nucleus, interactions with other adjacent transcriptional proteins, DNA methylation, etc. Similarly, line 263 should not use the term "affinity". The data shown in Supplementary Figure 4B suggests increased recruitment of Ets1 to proximal promoters of genes in adipocytes under conditions of cold-stress.

Sometimes, it is difficult to tell from the text or figure legends what tissue or cell type is being analyzed. For instance, it is difficult to tell the tissue/cell type in Figures 1F, 1G, 3D-E and 3G-I. What is "Lv", "Mus" and "Hypo" in Supplementary Fig. 2 and Supplementary Fig. 3A?

While Ets1 overexpression does lead to dramatic downregulation of mitochondrial proteins (Fig. 7A), the upregulation of these proteins in Ets1 KO cells is not very convincing (Fig. 7b). Likely, the downregulation seen in 7A is due to a loss of mitochondria, while the samples in 7B probably had similar or nearly similar numbers of mitochondria. Also, in 7F, the effects of M2 macrophage conditioned media are very weak.

The data in Figure 3A is very dramatic, but one wonders what the survival of EA+/- mice or EAKO mice would be?

The internal structure of the mitochondrial cristae seems significantly altered in Figure 7C. One wonders if a protein such as TMEM11, which controls cristae length and orientation and is involved in mitophagy, might be altered in expression (see PMID: 36795401).

Reviewer #3 (Remarks to the Author):

Wu et al. reported a set of data associated with the role of Ets1 in mediating the adipose tissue beige process in response to cold stress. Several questions and concerns as follow could challenge the conclusions of this study:

- 1) Body weight and fat weight after 6-OHDA treatment should be provided.
- 2) In Fig 1E-H, the control group without 6OHDA and clodronate liposome treatment should be included.
- 3) The data in Fig 1 are not sufficient to support the conclusion: "macrophages alone could induce adipose beigeing independent of SNS input" (lines 109-110, page 6). In Fig 1, a group treated with only clodronate liposomes (without 6OHDA) should be included to demonstrate the roles of macrophages.
- 4) Wrong labels: should be Fig 1J on line 121, Fig 1K on line 123, and Fig 1L on lines 128 and 131.
- 5) What could be the mechanisms underlying the larger adipocyte size in 6OHDA+CL (Fig 1H)?
- 6) Ets1 is among the DBPs in Fig 1K? From the ATAC-seq data, could any genes be regulated by Ets1? consistent with the findings from ChIPseq analysis?
- 7) The connection of sequencing data in Fig 1J and published dataset in Fig 1L and 1M&N is not well established, especially in demonstrating the interaction between macrophages and adipocytes.
- 8) Ets1 data in Fig 2C and 2D are not consistent. Ets1 expression in mature adipocytes (Fig 1M and Fig 2C and 2F) is very low or barely detectable, but Ets1 expression of 0day mature adipocyte-derived from iWAT in Fig 2D is robust.
- 9) To mimic in vivo conditions, authors should use CDM from adipose tissue macrophages for coculturing assays.
- 10) What could be the mechanisms underlying the larger adipocyte size in EA+/- mice (Fig 3C)?
- 11) Body weight and fat weight after overexpression of Ets1 should be presented.
- 12) The adipocyte size phenotypes shown in Fig 3C and 3F occurred in response to cold stress? or these phenotypes were formed because of Ets1 overexpression, even housed at room temperature.
- 13) In Suppl Fig 3A, fat pads were used. Since adipocyte-specific Ets1 knockout and other cells in fat pads, such as macrophages and endothelial cells, are not affected, why Ets1 was completely depleted in EAKO samples?
- 14) As mature adipocytes from iWAT of lean WT mice expressed a very low abundance of Ets1 and even less in response to cold stress, it is puzzling how ChIP-seq assays were done with these cells. It is possible that other cells such as macrophages or adipocyte precursor cells were bound with adipocytes were co-isolated from iWAT and contributed to Ets1 abundance.
- 15) References should be added for lines 274-275.
- 16) How is ASPC isolated for Fig 6A? However, the fig legend stated "SVF", then it should be SVF. Similarly, fig 6F is labeled with ADSC but fig legend states "SVF".
- 17) All histology or immunofluorescent images should be included with a scale bar.
- 18) What could the factor(s) in M2 CDM regulating Ets1 expression?
- 19) While the aim of this study focused on the regulation of M2 macrophages on adipose tissue beigeing, there is a limited amount of data supporting how macrophages mediate the adipocyte beigeing process. In addition, the IL4-induced BMDM (M2 macrophages) are not the same cells as adipose tissue macrophages.

In the response letter, the reviewer comments are laid out in **blue color**, while our point-to-point response is given in **red color**; The figure with a green background is the recently added data, while the figure with a blue background is the result already present in the original manuscript. In addition, in order to match the Figure/Panel numbers mentioned by the reviewer, the Figure numbers in this response letter still use the original ones, although some have been updated in the revised manuscript.

Reviewer #1 (Remarks to the Author):

In this study, authors identified that Ets1 is a key mediator of beige adipogenesis stimulated by M2 macrophages. They found that Ets1 is expressed in SVF but not in mature adipocytes. However, when using Ets1 adipocyte specific knockin and knockout, they found that Ets1 affected adipocyte specific mitochondrial density and oxidative function. While data are potentially important and interesting, additional studies are needed in order to conclusively test the mediatory roles of Ets1.

The following are major comments:

1. Since Ets1 is only expressed in SVF, not mature adipocytes (Fig. 2C), why did adipocyte specific knockout and knockin were selected (a)? This question is partially alleviated by the Fig. 2D, where appeared that Ets1 is expressed during early adipogenesis (b). However, there is still a question about the biological effects of Ets1, whether it is mainly through promoting beige adipogenic commitment and differentiation, or simply through enhancing mitochondrial density and function (c)?

Response:

Thank you for the valuable comment.

(a) Although relatively low, Ets1 is indeed expressed in mature adipocytes, and the expression level is correlated with stimulations such as cold stress or diet induced obesity. Therefore, we chose adipoq-cre as the tool mice. We apologize for the possible misleading of immunoblot data in Fig.2C. In fact, as exposure increase, Ets1 expression can also be detected in mature adipocytes (below). Therefore, in Line 163-164 of the original manuscript, we mentioned “Ets1 was mainly expressed in the ASPC-enriched SVF cells”. Our data in Fig.2E, and Suppl-Fig.1B (below) supports the hypothesis that Ets1 is low expressed in mature adipocytes, but not completely absent. In addition, the low expression of Ets1 in mature adipocytes is not due to the consistent low level of Ets1 in all adipocytes, but rather to the relatively low portion of cells expresses Ets1 (Suppl-Fig.1B). Therefore, we selected the adipoq-cre to either force Ets1 expression in all adipocyte, or to block the expression of Ets1 in the small portion of adipocytes that originally expressed Ets1.

Figure Above:

Fig. 2C: Western blot of *Ets1* in mature adipocytes (adipo) and the stromal vascular fraction (SVF) of iWAT and eWAT (n=2).

Fig. 2E: SVFs separated from wild type C57 were differentiated into beige adipocytes. At indicated time point after differentiation, protein level of *Ets1* were analyzed by western blot (n=2).

Supp-Fig. 1B: UMAP projection of only the ASPC and mature adipocytes of both chow diet and high fat diet fed mice (left). The expression of *Ets1* across these cells were profiled (right). A published single cell transcription sequencing dataset (GSE176171) were used.

New data A: long exposure image of Fig.2C.

(b) Besides, we need to point out that Fig.2E, rather than Fig.2D, provides a correlation study between *Ets1* expression and beige adipogenesis, as Fig.2E used SVFs derived beige adipocytes whereas Fig.2D uses total iWAT lysate. We conducted a more detailed study on Fig.2D and data suggested that *Ets1* was indeed expressed in mature adipocyte (Panel below, A).

Figure Above: Western blot showing the protein levels of *Ets1* in iWAT fat pad (Fig.2D), in mature adipocyte (A) or in SVF (B) derived from iWAT of mice pretreated with cold stress for indicate time.

(c) Based on current data, we can confirm that *Ets1* has an impact on the content and function of mitochondria, which will definitely lead to changes in the beige adipose function. However, we do not yet clear whether *Ets1* could promote beige adipogenic commitment and differentiation. We cautiously speculate that *Ets1* may also have an impact on beige lineage determination, which is independent of its effect on mitochondria. The reason is that compared to beige fat, knockout or over-expression of *Ets1* has almost no effect on the activity of brown fat, which also enriched in mitochondria. The biological effect of *Ets1* on the adipocyte lineage determination and development, which hardly can be conducted using the current adipoq-cre tool mice, will be analyzed in future studies.

2. 6OHDA was used to induce sympathetic nerve ablation. However, 6OHDA induces broad changes in neurons, not only sympathetic nerves. To be conclusive, additional method such as physical methods may be needed.

Response:

This is a good question. 6OHDA may indeed have non-specific effects, however, due to the control group setting, the potential non-specific effects will not largely affect the conclusions of this study. As the referee suggested, we tried to performed surgical SNS ablation on mice, but it seems that the surgery was not successful—we did not observe a significant reduction of NE level in the fat pad of the denervated mice.

—As for the nonspecific effect of 6OHDA

In the current study, mice injected with 6-OHDA alone were used as the control group, while mice injected with 6-OHDA plus clodronate liposome were treated as the model group. Both groups of mice were injected with 6-OHDA, so even if there is non-specific effect, this effect can be effectively eliminated via comparison between the two groups.

—As for the SNS resection

There have been minimal studies on adipose tissue sympathetic nerve resection in mice, but the corresponding method in golden hamsters is relatively mature (PMID: 24480348). We followed the hamster-proven method and conducted SNS resection in mice (Panel below, A), and the specific protocol are as follows.

Unfortunately, we found that the NE content in the iWAT tissue of denervated mice did not significantly decrease (Panel below, B), indicating that surgery could not effectively remove the sympathetic nerves that governs this fat pad. We are currently unclear whether this is due to the different nerve localization between mice and golden hamsters, or it is due to the incorrect surgical procedure. Hence, surgical resection from the upstream branches (even ganglion) of the sympathetic nervous system is needed. We will optimize the surgery and analyze the effect of clearing macrophages under this model in future research.

Figure above:

A: left, the nerves innervating iWAT were identified (black arrow); right, the nerve were cut and removed (red arrow). B: 14 days later, the iWAT were separated and used for NE determination.

Mice iWAT SNS surgical resection protocol:

(1) Skin incision in mice under isoflurane gas anesthesia. Wipe area with 75% ethanol- soaked sterile gauze followed by additional application of Betadine to the area. Make an incision in skin dorsally on the flank from a point near the tail and lateral to the spinal column. Continue the incision rostrally along the dorsum adjacent to the spinal column to a point just rostral to the hind limb, then laterally and ventrally to a point ~1 cm from the ventral midline.

(2) Denervation. Tap one drops of 1% toluidine blue directly onto fat pad to better visualizing the nerves innervating iWAT. Nerves identified at $\times 4$ magnification that terminate in the pad (Panel above, A left) should be cut in two or more locations, followed by removing the nerve sections between the cuts to assure denervation and the unlikely possibility of nerve regrowth (Panel above, A right, red arrow pointed to the removed nerve). The sham procedure includes fat pad manipulation and nerve identification, with exception that the nerves are left intact.

(3) Closed the skin incision by suture, and applied nitrofurazone powder to the wound surface to minimize the risk of bacterial infection. After awakening, the experimental animals were raised for 14 days to wait for the complete disappearance of existing NE in adipose tissue. The fat pads were then separated and used to measure NE content.

3. Similarly, clodronate liposome was used for locally ablate the macrophages, which is insufficient. The study argues that M2 macrophages, not M1 is responsible. Therefore, additional testing in M1 and M2 transition and their effects on Ets1 expression and beige adipogenesis is needed.

Response:

This is a good question. We performed the reviewer mentioned assay, and result suggested that M1 macrophage raised adipocyte Ets1 expression, blocked the beiging process, while M2 macrophage showed the opposite effect.

As the referee suggested, we determined the regulatory effect of M1 and M2 macrophage on Ets1, as well as on beige adipogenesis. Our results revealed a clear regulatory effect of macrophages on the expression of Ets1 in adipocytes: adipocytes treated with M1 macrophages conditional medium (CDM) showed enhanced Ets1 expression (A-B, below), which is exactly opposite to the effect of M2 macrophages CDM (Fig.2 K-M). In addition, overexpression of Ets1 can significantly exacerbate the inhibitory effect of M1 macrophages on beige adipogenesis (A-B, below).

Figure above:

Fig.2K-M: SVFs separated from iWAT of wild type C57 mice were cultured *in vitro*. The plate-fully-covered cells were infected with indicate adenovirus for 6h, then induced differentiation towards beige adipocytes. 5 days later, the cells were treated with 50% M2 CDM for 24h.

K: Western blot showing the protein levels of Ets1 and thermogenic markers (n=2).

L: Quantitative PCR showing the mRNA levels of Ets1 and thermogenic markers (n=3).

M: Oxygen consumption rate (OCR) measured at Day 8 (n=6).

NEW DATA:

A-B: SVFs separated from iWAT of wild type C57 mice were cultured *in vitro*. The plate-fully-covered cells were infected with indicate adenovirus for 6h, then induced differentiation towards beige adipocytes. 5 days later, the cells were treated with 50% M1 CDM for 24h.

A: Western blot showing the protein levels of Ets1 and thermogenic markers (n=2).

B: Quantitative PCR showing the mRNA levels of Ets1 and thermogenic markers (n=4).

To further clarify the biological function of adipocyte Ets1 in responding to M2 macrophages stimuli as well as promoting beige adipose formation *in vivo*, we recently performed macrophage adoption assay, either on control or adipocyte Ets1 knock-in mice. Our results suggested that while M2 macrophage adoption raised the mitochondria content (A, below), promotes the cold induced beige adipogenesis (B, below), up-regulated the body temperature (C, below), enhanced the thermogenic gene expression (D, below), forced Ets1 expression significantly reversed all these effects (A-D, below).

Figure above:

A-D: M2 BMDMs were adopted into each side of iWAT, at day 1, 3 and 5 respectively, ensuring that the amount of each injection was 10^6 . The control group was injected with the same volume of PBS. The mice were then given 6-OHDA injections at day 5, 7, and 9 to operate sympathectomy. At day 10, mice were cold stressed at 10°C for 6 h and then 4°C for 4 days, after which samples were collected. $N=6$ per group.

A: Western blot analysis of the level of OXPHOS, Ets1, and Ucp1.

B: General morphology of iWAT (left) and H&E staining of iWAT (right).

C: Rectal temperature measured during cold stress.

D: qPCR testing Ets1, macrophage marker F4/80, and thermogenic fat marker genes.

The above assay suggested that the adipocytes response to the binary M1/M2 polarized macrophage and alter the expression level of Ets1, via which then mediated the macrophages stimulation and regulated the beige adipogenesis. However, considering that (1) macrophages in adipose tissue of lean mice are mainly M2 polarized (only $9.3\% \pm 7.5\%$ were M1 like, PMID: 17200717); (2) the beige adipogenesis promoting effect are mediated by M2 macrophages (PMID: 24906147, PMID: 34782792). Therefore, we reasonably speculate that injecting Chlorophosphate liposomes in cold-stressed lean mice mainly affects M2 macrophages, and focusing on M2 macrophages-Ets1-beige adipogenesis is of great research value.

4. In figures 4 to 7, Ucp-1 was not analyzed. However, Ucp-1 expression is a critical marker of being. Therefore, related analyses should be added.

Response:

As the referee suggested, we analyzed Ucp1 in Figure 4-7.

For Fig.4: In the iWAT of diet-induced obese (DIO) mice, the Ucp1 was undetectable in WB (A, below) and relatively low ($Ct>30$) in qPCR (B, below). While in the BAT of these DIO mice, blocking Ets1 slightly increased the expression of Ucp1 (C-D, Below), consistent with the multi-cellular H&E staining results (Fig.4D).

Figure above:
 After 14 weeks of HFD feeding, mature adipocyte separated from iWAT (A-B) or BAT (C-D) of either EAKO or Flox mice were collected, and used to detecting the expression of Ucp1 and Pgc1, both in protein (A, C) and in mRNA (B, D) levels.

Fig.4D: H&E staining of iWAT and BAT sections.

For Fig.5: The potential binding peaks of Ets1 on Ucp1 were profiled, and data suggested that Ets1 showed no obvious binding peak to the promoter of Ucp1 (A, below). Namely, the regulatory effect of Ets1 on Ucp1 expression was not rely on a direct transcriptional repression.

Figure above: The IGV tool was used to visualize the binding peaks of Ets1 on the promoter region of Ucp1.

For Fig.6: As the referee suggested, we analyzed the expression of Ucp1 in Fig.6A. Result showed that Ets1 overexpression reduced the Ucp1 level at day 4 and day 6 (A, below). For Fig.6B-C, the samples used are the same as those in Fig.4 above, and the expression of Ucp1 has been detected (see “For Fig.4” in this response). For other panels in this figure, the used sample were either 3T3-L1 (Fig.6E) or un-differentiated SVF cells (Fig.6F), and Ucp1 were low expressed and un-detectable by WB in these cells.

Figure above:
 A: qPCR testing of the mRNA levels of Ucp1. SVFs were separated from iWAT of wild type C57 mice were infected with a corresponding adenovirus. Beige adipogenesis was induced 24 h later (deemed day 0). n=3 per group.
 Fig.6 E: Western blot analysis of autophagy in 3T3L1 cells (n=2). Cells were either infected with ad-shEts1 or ad-scramble adenovirus; 48 h later, the medium was changed to low FBS (0.5%, starvation) or fresh normal FBS (10%), and the cells were cultured for another 24 h.

Fig.6 F: Western blot testing the protein levels of *Pink1*, *Prkn*, and *Ets1* (n=3). SVFs were infected with or without *ad-Ets1* adenovirus and analyzed 48 h later. Before sample collection, the cells were treated with 0 or 20 μ M CCCP for 3 h to activate mitophagy.

For Fig.7: In this figure, the presented data focusing on the regulatory effect of *Ets1* on mitochondria, while the corresponding phenotypic data of beige adipogenesis, such as *Ucp1* expression, have been shown elsewhere. For example, the *Ucp1* of panel Fig.7F have been analyzed in Fig.2K. In the recently completed M2 macrophage adoption assay, the expression of *Ucp1* were analyzed and added to revised manuscript (A-B, below).

Figure above:

Fig.2K: Western blot showing the protein levels of *Ets1* and thermogenic markers (n=2). SVFs separated from iWAT of wild type C57 mice were cultured *in vitro*. The plate-fully-covered cells were infected with indicate adenovirus for 6h, then induced differentiation towards beige adipocytes. 5 days later, the cells were treated with 50% M2 CDM for 24h.

New data A-B: Quantitative PCR (A) and Western blot (B) analysis of *Ucp1*, and other thermogenic expression. M2 BMDMs were adopted into each side of iWAT, at day 1, 3 and 5 respectively, ensuring that the amount of each injection was 10^6 . The control group was injected with the same volume of PBS. The mice were then given 6-OHDA injections at day 5, 7, and 9 to operate sympathectomy. At day 10, mice were cold stressed at 10 $^{\circ}$ C for 6 h and then 4 $^{\circ}$ C for 4 days, after which samples were collected. N=6 per group.

5. Just for easy to read, in figures, abbreviations, especially unstandardized, should be minimized and treatments/experimental design can be briefly described in the figure legends.

Response:

Thank you for the valuable comment. In the original manuscript, we used some abbreviations to make the figure concise. This can indeed lead to a certain degree of reading difficulty. We now follow the reviewer's suggestion and try to use the full name in the figure as much as possible (e.g., CL in Fig.1 has been changed to Clod Lipo).

6. I could not find the Method Section and, thus, cannot evaluate experimental procedures.

Response:

Sorry for the mistake. Considering the length of the manuscript, we split the Method Section into a supplementary document (named Material and Method) and uploaded it separately. We are sorry for this inconvenience and hopefully you can find this section in the revision, otherwise please let me know.

The following are detailed comments:

Line 60: The reference 14 is a review paper, original study is needed here.

Response:

Thank you. Although the name may look like a review, the mentioned Ref.14 is indeed a research paper. This paper is a comparative study published in Am J Physiol, please use the pubmed like below for quick glimpses of the study.

Dietary obesity and neonatal sympathectomy. II. Thermoregulation and brown adipose metabolism - PubMed (nih.gov)

Line 83: Reference is needed after M2 macrophage stimuli.

Response:

Thank you for the comment. The mentioned Line83 (complete sentence excerpt below) is not a summary of a previous research, but a discovery of the current study. In order to avoid misleading caused by improper writing, we have rewritten this sentence as following.

Original: Multiple sequence-based screening identified Ets1, a transcription factor that plays a fundamental role in maintaining glucolipid homeostasis [31, 32], was a mediator of M2 macrophage stimuli.

Changed to: We performed multiple sequence-based screening and identified Ets1, a transcription factor that plays a fundamental role in maintaining glucolipid homeostasis [31, 32], serve as a mediator of M2 macrophage stimuli.

Line 91: How about the nonspecific effects of 6OHDA?

Response:

This is a good question. In our study, 6OHDA may indeed have non-specific effects, such as affecting other organs, but this will not largely affect the subsequent data analysis and conclusions of this experiment.

6-hydroxydopa (6-OHDA), the hydroxylated derivative of dopamine, is widely acknowledged as an efficient pharmacological tool for sympathetic nervous system denervation. Due to the structural similarity to dopamine, 6-OHDA can be uptake by synapse through dopamine transporters (Slc6a3) or norepinephrine transporters (Slc6a2), and then damage neurons through various mechanisms (such as oxidative stress). The effect of 6OHDA in blocking SNS is reliable and clear, which has been confirmed by numerous studies such as in adipose (PMID: 28918935), in liver (PMID: 12939593), in stomach (PMID: 2501024), and in immune system (PMID: 19286971 and PMID: 11498048).

In the current study, we performed 6-OHDA-based chemical SNS denervation, and the blocking efficiency in the fat pad were further confirmed by detecting NE and tyrosine hydroxylase expression. As for the mentioned non-specific effects of 6-OHDA, we cannot completely rule out the possible uptake of 6-OHDA by other tissues, as well as by adipose catecholaminergic nerve terminals other than SNS. But the control group setting in this study can efficiently exclude the impact of the potential non-specific effects. In this study, mice injected with 6-OHDA alone were used as the control group,

while mice injected with 6-OHDA plus clodronate liposome were treated as the model group. Both groups of mice were injected with 6-OHDA, so even if there is non-specific effect, this effect can be effectively eliminated via comparison between the two groups.

Line 98: Clodronate liposome ablation removes M2 macrophage specifically? How about M1 macrophage? An additional method which can alter the polarization of macrophages is needed.

Response: Thank you for the question. Chlorophosphate liposomes target all macrophages non-specifically, therefore, M1 macrophage can also be affected. As the referee's suggestion, we analyzed the regulatory effect of M1 macrophages on adipocytes Ets1 expression level, as well as on beige adipogenesis. Detailed results can be found in the previous answer to major question 3, which is similar to this question.

Line 191: EA+/, Ets1ki/ki×Adipoq-cre can be changed to Ets1ki/ki×Adipoq-cre; abbreviated as EA+/+.

Response:

Thank you for the comment. Changed according to the referee's suggestion.

Line 245: Changes in being need to be analyzed.

Response:

As the referee's suggestion, the expression level of Ucp1, together with beige marker Pgc1 α , were determined. This question is also answered in response to question 4 in major concern

In the iWAT of diet-induced obese (DIO) mice, blocking Ets1 leads to an increase in the expression of Pgc1 α , however, in HFD mice, the Ucp1 was undetectable in WB (A, below) and relatively low (Ct>30) in qPCR (B, Below). While in the BAT of these DIO mice, blocking Ets1 significantly increased the expression of both Pgc1 α and Ucp1 (C-D, Below), consistent with the multi-cellular H&E staining results (Fig.4D).

Figure above:

New data A-D: After 14 weeks of HFD feeding, mature adipocyte separated from iWAT (A-B) or BAT (C-D) of either EAKO or Flox mice were collected, and used to detecting the expression of Ucp1 and Pgc1, both in protein (A, C) and in mRNA (B, D) levels.

Fig.4D: H&E staining of iWAT and BAT sections.

Line 251: Why only WT mice? This should be done in Ets1 knockin and knockout mice.

Response:

This is a good question. While conducting ChIP seq-based mechanism study (Line 251), it is more reasonable to discover the downstream target of endogenous Ets1 under different physiological states (such as normal temperature/cold stress). The reasons are:

1) Overexpression of Ets1 may lead to non-specific binding on chromatin.

We are concerned that overexpression of target proteins, which leads to much higher level beyond physiological level, may lead to potential non-specific binding and the subsequent sequencing will further amplify this noise.

2) The expression of Ets1 decreases under cold stimuli, thus comparing the differential Ets1 ChIP-seq signals with/without cold stress is meaningful.

We performed Ets1 ChIP-seq under different physiological states: the mice were treated with/without cold stress, which will change the endogenous expression level of Ets1. Therefore, by comparing the differential sequencing signals between room temperature and cold stress, which contains high and low level of Ets1 respectively, we aim to explore meaningful Ets1 downstream targets in the real state.

3) In ChIP-seq assay, analyzing the binding signals of endogenous proteins under different physiological states is a widely received research method. Attached below are two published studies using similar method for ChIP-seq.

Line 293: It is claimed that Ets1 works as a classic transcription factor suppressing mitochondrial coding genes. If this is the case, it is necessary to analyze additional tissues enriched with mitochondrial, such as muscle, liver, etc.

Response:

This is a good question. The negative regulatory effect of Ets1 on mitochondrial coding genes were confirmed in muscle and liver cells.

In primary hepatocytes, forced expression of Ets1 significantly reduced the expression of mitochondrial complexes, both in the protein (A, below) and in the mRNA levels (B, below). In the muscle cell line C2C12, overexpression of Ets1 by virus was not as effective as hepatocyte. However, it can still be seen that in C2C12 cells, Ets1 reduced the mRNA level of mitochondrial coding genes (C, below), and inhibits parts of the mitochondrial complex (mainly complex I) (A, below).

Figure above: Primary hepatocyte (A, B), as well as C2C12 muscle cell (A, C), were infected with/without Ets1 overexpression adenovirus. 48h later, cells were harvest to detecting the expression of mitochondria complex expression, either in protein level (A) or in mRNA level (B-C).

Figure 1D: Why did F4/80 differ so dramatically due to CL?

Response:

Thank you for the question. Clodronate (CL) is a strong macrophage scavenger, and systemic or local injection can significantly reduce macrophage levels in target organs, thereby reducing F4/80 mRNA levels. Several other studies analyzing the mRNA of F4/80 after Clodronate mediated clearance showed similar data (shown below), consistent with the results of this study.

PMID: 19875703(Fig.4H)

PMID: 24349208 (Fig.3B,D)

Figure 1N: The meaning of horizontal changes needs to be labeled in panel N. Pseudotime started as mature adipocytes and ends in ASPC?

Response:

Thank you for the question. Actually, the pseudotime start as ASPC (left) and ends in mature adipocytes (right). During this transition, it can be clearly seen that the expression of ASPC marker gene, such as DPP4, or genes enriched in ASPC, such as Fstl1 and Egfr, gradually decrease; while the expression of mature adipocytes marker genes, such as Adipoq and Pnpla3, significantly increased.

The horizontal changes have been labeled as the referee's suggestion.

Figure above: Pseudotime differential trajectory analysis of ASPC and mature adipocytes.

Figure 2C: Panel C showed that the Ets1 is mainly expressed in SVF, not adipocytes. Then why was Adiponectin-Cre was used? Why Ets1 inhibits mitochondrial biogenesis in mature adipocytes if it is not expressed?

Response:

Thank you for the question.

Actually, Ets1 is low expressed in mature adipocytes, but not completely absent. In addition, the low expression of Ets1 in mature adipocytes is not due to the consistent low level of Ets1 in all adipocytes, but rather to the relatively low portion of cells expresses Ets1. The question is basically the same as major concern 1, and answered in detail above.

Figure 2: Could an additional approach which can inhibit M2 macrophage transition be used to further verify? N=2 is too low.

Response:

Thank you for the comment.

We determined the regulatory effect of M1 macrophage on Ets1, as well as on beige adipogenesis. The data and related analysis have been presented in detail in the answer of major concern 3 above.

Figure 4J-M: How about heat production?

Response:

The heat production data, which were shown in the supplementary Fig.3 (H-I), suggested that EAKO mice generates more heat than the control littermates (Supp-Fig. 3H-I, below). The Oxygen consumption (Fig. 4J-M), the heat production (Supp-Fig. 3H-I), together with VCO₂ generation (Supp-Fig. 3F-G), showed similar changes, thus we displayed the last two in supplementary data. We are sorry for the inconvenience and hopefully you can find this data in the revision.

Figure above: Heat generation (H) and the calculated AUC (I) of EAKO and Flox mice, after 14 weeks of HFD feeding.

Figure 5D: It will be more informative if the length of promoter region can be indicated, including TSS.

Response:

Thank you for the comment. In the luciferase assay, the length of all the used promoter were 1100bp, that is from -1000 to +100. We followed the reviewer's suggestion and labeled the "-1000", "TSS", and "+100" in the revised figure, as shown below.

Figure above: Promoter plasmid construction and the relative luciferase activity. The "-1000", "TSS", and "+100" are shown in the revised figure.

Figure 5F: How about UCP-1 and beige related genes?

Figure 5G: Why not UCP-1, the most important gene for beige adipogenesis?

Response to the above two questions:

Thank you for the question. Ucp1 is not a direct transcriptional target of Ets1, therefore it was not analyzed in luciferase assay.

The potential binding peaks of Ets1 on Ucp1 were profiled using Ets1 ChIP-seq data, and result suggested that Ets1 showed no obvious binding peak to the promoter of

Ucp1 (Below, A). Namely, the regulatory effect of Ets1 on Ucp1 expression was not rely on a direct transcriptional repression.

Figure above: The IGV tool was used to visualize the binding peaks of Ets1 on the promoter region of Ucp1.

Figure 5H-I: Why mature adipocytes?

Response:

This is a good question. Only the mature adipocytes were used for library building, in order to exclude the potential sequencing noise cause by none-adipocyte in fat tissue.

Although adipocytes dominate the cellular composition of adipose tissue (in a healthy state), but strictly speaking, adipose tissue is a pool of cells mixed with various other cells, such as macrophages and endothelial cells. As the Ets1 knock-out and knock-in were specifically localized to adipocytes, we attempted to exclude the possible interference of sequencing data caused by other cells in Figure 5H-I. Therefore, only mature adipocytes were used for library construction and sequencing.

Figure 7: This study appears not related to M2 macrophage? A brief description of experimental design in figure legend can help.

Response:

Sorry for the misleading. In Figure 7, panel F-G, and the newly performed M2 adoption assay (H-J), were M2 related, while panel A-E were simple related to Ets1, not M2 macrophage. We briefly described the experimental design of these two parts in the revised figure legend.

Figure 7F-G: Should also be separated from EA+/+ and EAKO mice?

Response:

Thank you for the question. To better mimic the *in vivo* condition, we conducted M2 macrophage adoption assay, using EA+/+ and control mice, and obtained consistent data.

Compared to the reviewer mentioned *in vitro* M2 macrophage CDM culture assay, macrophage adoption represents a better method that discover the *in vivo* biological function of adipocyte Ets1 in responding to M2 macrophages and in regulating beige adipogenesis, in a complex tissue environment under physiological state. The results of M2 macrophage adoption assay have been thoroughly discussed in response to major concern 1, and also added to Fig.7 in the revised manuscript.

Reviewer #2 (Remarks to the Author):

The manuscript by Wu et al describes the very interesting finding that M2 macrophages can promote thermogenic beige adipogenesis independently of their effect on the sympathetic nervous system and that this effect is dependent on suppressing the expression of transcription factor Ets1 in adipocytes (particularly in adipose stem and progenitor cells (ASPC)). The data presented are generally convincing and there is a dramatic effect loss of Ets1 in adipocytes resulting in the complete loss of cold tolerance in mice.

1. The results are overall interesting, but it would be nice to know how M2 macrophages affect Ets1 levels in adipocytes. Does one of the cytokines released by M2 cells, such as IL-6, IL-1 β or TNF- α , affect Ets1 levels?

Response:

This is a good question! Our data suggested that the regulatory effect of M2 macrophage on adipocyte Ets1 was not mediated by cytokines. The mediator should be small molecule that less than 10KD, but the specific molecule is currently unclear.

Actually, we already analyzed the regulatory effects of M1/M2 macrophage derived cytokines on the expression of Ets1 in adipocytes. Our data suggested that IL-10 (mainly secreted by m2 macrophages), as well as IL-6, IL-1b, and TNF α (mainly secreted by m1 macrophages) treatment could not significantly change the expression of Ets1(below, A).

Recently, we separated the M2 CDM by ultrafiltration tubes, which can distinguish the components in the medium by molecular weight, and then treated the SVF cells with these separated components. We found that even with the smallest ultrafiltration tube, which only allow the passage of molecules less than 10kd, the filtrated solution could still significantly inhibit the expression of Ets1 (below, B). This highly suggests that the inhibitory effect of macrophages on Ets1 is unlikely to be exerted through large molecular weight components, such as secreted proteins or exosomes, and the possibility of acting through inflammatory cytokines or chemokines is also relatively small. We speculate that it is possible that certain small metabolites secreted by M2 macrophages inhibit the expression of Ets1 in adipocyte. Further research (such as metabolomics) is still needed, to discover the mediator and clarify the underlying mechanism.

Figure above:

A: Western blot showing the expression of Ets1 in 3T3-L1, which were pretreated with corresponding cytokines for 4h.

B: Western blot showing the expression of Ets1 in SVF cells, which were pretreated with M2 CDM for 4h. Control: 0% M2 CDM; Total: 50% M2 CDM; <10 KD and >10KD: M2 CDM were separated via ultrafiltration tubes, the media containing molecules < 10KD, and > 10KD were collected, and used to treat SVF cells at 50%.

2. In Figure 1E, what is the effect of macrophage depletion alone as opposed to macrophage depletion in the context of SNS ablation?

Response:

Thank you for the question. Compared with macrophage depletion plus SNS inhibition, the macrophage depletion alone can also inhibit adipocyte browning and thermogenesis, but the effect is much weaker.

The mentioned assay in Fig.1 was repeated with additional control groups, such as the reviewer mentioned macrophage depletion alone group, as well as the reviewer 3 suggested control group that without 6OHDA or clodronate liposome treatment. Compared with macrophage depletion plus SNS ablation group, the macrophage depletion alone showed much smaller effect on blocking beige adipogenesis, manifested as higher body temperature (below, A), more expression of thermogenic genes (below, B-C), and corresponding morphological changes (below, D).

Figure above:

A-D: Wild type C57/BL6j mice received intraperitoneal injection of 6-OHDA (100 mg/kg) or vehical, on day-5, -3, and -1. At day 0, mice were subcutaneously injected 4 times (2 for each side) with 50 ul clodronate liposomes or control liposomes into iWAT. The liposomes injection was repeated once at day 2. All the four group of sham (n=3), 6OHDA alone (n=3), Clod Lipo alone (n=3), and 6OHDA+Clod Lipo (n=3) were cold stressed for 4 days, started from Day 0.

A: Rectal temperature measured after a 4-day cold stress.

B: Western blot analyzing the protein level of thermogenic marker *Pgc1a* and *Ucp1*.
 C: Quantitative PCR analyzing the mRNA level of beige adipose marker genes.
 D: General morphology of iWAT (up) and H&E staining of iWAT (down).

3. How can you tell macrophage depletion by clodronate is limited to iWAT? It is not clear that the liposomes will only be found in iWAT and not spread systemically.

Response:

This is a good question. We tested visceral fat, as well as monocyte within the circulating blood, and both the results can support the clearance of macrophage is limited to the iWAT.

After clodronate treatment, the iWAT, as well as eWAT, was collected and used to analyzing the expression of F4/80, marker gene of macrophage. Quantitative PCR suggested that the liposomes injected within the iWAT, could not affect the eWAT (below, A-B). As for the mentioned systemically spread of liposomes, previous study showed that mice injected with clodronate via tail vein showed reduced monocytes within blood (PMID:24354415, below). Hence, we performed blood cell analysis of mice pretreated with/without CL, and found that CL treatment showed no effect on the count of monocytes and neutrophils (below, C-D). Taken together, our data suggested that the clodronate treatment is limited to iWAT, and not spread systemically.

Figure above: After 4-days cold stress, the four group of sham (n=3), 6OHDA alone (n=3), Clod Lipo alone (n=3), and 6OHDA+Clod Lipo (n=3) were analyzed.

A-B: Quantitative PCR testing the expression of macrophage marker F4/80 in iWAT (A) and eWAT (B).
 C-D: Monocyte number (C) and neutrophil cell number (D) in circulating blood.

4. There is a sentence saying “Consistently, the effects of Ets1 overexpression on BAT thermogenic activation were relatively small” in lines 216-217. However, the histology of the BAT with Ets1 overexpression in figure 3F seems to be significantly different than the control. Perhaps since BAT has more mitochondria to begin with, this tissue can withstand better the effects of Ets1 overexpression? Can the function of the BAT be more directly tested?

Response:

This is a good question. We recently conducted multiple assays, such as Ucp1 IHC and sea horse assay, and results suggested that there is indeed no significant change in the BAT activity.

Compared with H&E staining, which is merely a morphological level detection method and cannot directly measure thermogenic ability, analyzing the content and localization of Ucp1 via immunohistochemistry staining could present more direct and convincing results of the BAT activity. In IHC assay, we found very interesting results: in all 6 mice (3 controls, 3 ki), although Ets1 overexpression significantly increased the size of lipid droplets in BAT, the Ucp1 IHC staining was remain comparable (below, A). Therefore, we believe that WB data of Ucp1 (Fig.3G) is accurate, that Ets1 KI mice have little effect on Ucp1 levels.

In order to further analyze the activity of BAT, the SVF cells were separated from BAT of Ets1 knock-in or control mice, and differentiated into brown adipocyte and then analyzed by sea horse assay. Forced expression of Ets1 showed no effect on either the basal or maximal respiration (below, B). Namely, the effect of Ets1 overexpression on BAT activity was relatively small.

Figure above:
 A: Immunohistochemistry (IHC) staining of Ucp1 in BAT separated from homozygous Ets1 knock-in and control mice, that pre-injected with CI316243 for 7 days.
 B: Sea horse assay of differentiated brown adipocyte. SVF cell from BAT of EC+/+ and EA+/+ were cultured and differentiated for 7 days, before analysis.
 Fig.3F: Hematoxylin and eosin (H&E) staining of iWAT and BAT from same mice mentioned in A.
 G: Western blot assay analyzing Ucp1 and Pgc1α in iWAT and BAT from same mice mentioned in A.

5. Can the authors speculate why EAKO mice show increased locomotor activity?

Response:

This is a good question. We speculate that the increased motor ability of EAKO mice may be secondary to the effect of weight loss, rather than a direct impact on the muscle.

In our study, under the HFD-induced obese state, the adipocyte Ets1 knockout mice showed increased locomotor activity (Fig.4K), which we found very interesting and decide to present it in the figure. Though we are currently unable to determine the underlying mechanism, but we speculate this phenotype could be attributed to two reasons. (1) The changed expression of Ets1 in adipocytes affects the muscle function through a certain way, such as secreting adipokines. (2) Due to the overall weight loss of EAKO mice (Fig.4A), the thinner body leads to stronger locomotor activity.

Interestingly, we did not discover significant changes in locomotor activity in lean EAKO mice fed with normal chow diet, in comparison with corresponding lean control littermates (below, A). This indicates that blocking Ets1 in adipocytes do not directly affect physical activity, at least in the healthy state. Therefore, based on the current data, we speculate that the increased motor ability of EAKO mice may be secondary to the effect of weight loss, rather than a direct impact on the muscle. Further research is still needed, such as muscle morphology and electrophysiology examination of the EAKO mice, to fully reveal the potential fat-muscle regulatory pathway and the role of Ets1 in this circuit.

Figure above:
 Fig.4K: Daily locomotor activity of HFD-fed Flox and EAKO mice.
 Fig.4A: Body weight curves during HFD feeding.
 New data A: Daily locomotor activity of normal chow diet-fed Flox and EAKO mice.

6. The sentence in lines 256-257 stating that “cold stress significantly enhanced binding affinity of Ets1” should be revised. The CUT&TAG assay used doesn’t assess transcription factor affinity for DNA. Instead, it is assessing overall level of binding of Ets1 to DNA, which can be affected by multiple parameters including increased concentrations of Ets1 in the nucleus, interactions with other adjacent transcriptional proteins, DNA methylation, etc. Similarly, line 263 should not use the term “affinity”. The data shown in Supplementary Figure 4B suggests increased recruitment of Ets1 to proximal promoters of genes in adipocytes under conditions of cold-stress.

Response:

Thank you for the valuable suggestion.

We sincerely apologize for the incorrect wording and would greatly appreciate your valuable feedback on this flaw. In the previously submitted manuscript, we misused the word "affinity" to address the ChIP-seq data, which actually measures occupancy. As the referee mentioned, changes in affinity can change occupancy, but so can many other things, such as intracellular content of the target DNA binding protein. We have thoroughly read the entire manuscript and made modifications to the words and phrases where affinity was used to incorrectly describe ChIP-seq, and would like to thank you again for this valuable suggestion.

7. Sometimes, it is difficult to tell from the text or figure legends what tissue or cell type is being analyzed. For instance, it is difficult to tell the tissue/cell type in Figures 1F, 1G, 3D-E and 3G-I. What is "Lv", "Mus" and "Hypo" in Supplementary Fig. 2 and Supplementary Fig. 3A?

Response:

Thank you for the comment. We apologize for the reading difficulties caused by unclear labeling.

In Fig.1, due to limited space, we used "CL" instead of "Clodronate Liposomes".

In Fig.1 and Fig.3, we labeled the tissue/cell type in each panel, to make it easy to follow.

In supp-Fig.2 and supp-Fig.3A, the "Lv", "Mus" and "Hypo" means "Liver", "muscle", and "hypothalamus", respectively.

In order to minimize the understanding difficulties caused by abbreviations as much as possible, we have tried to use the full name (for e.g.: using Liver instead of Lv) in the revised manuscript. For overly long words, using the full name in the figure does not have enough space, so we have used a more appropriate abbreviation (for e.g.: using Clod. Lipo. instead of CL).

We once again thank you for your suggestions and contributions, and hope that the revised manuscript can improve your reading experience.

8. While Ets1 overexpression does lead to dramatic downregulation of mitochondrial proteins (Fig. 7A), (a) the upregulation of these proteins in Ets1 KO cells is not very convincing (Fig. 7b). Likely, the downregulation seen in 7A is due to a loss of mitochondria, while the samples in 7B probably had similar or nearly similar numbers of mitochondria. (b) Also, in 7F, the effects of M2 macrophage conditioned media are very weak.

Response:

Thank you for the comment. In Fig.7b, we conducted grayscale analysis to confirm

that the changes are significant. In Fig.7f, the effect was indeed very weak, which could be due to the relatively short timing of CDM treatment. We recently conducted *in vivo* M2 macrophage adoption assay to replace the Fig.7F, and to confirm the promoting effect of M2 macrophage on mitochondria content.

(a) Compared to Fig.7A, we noticed that HFD feeding leads to a relatively high level of Complex II in Fig.7B. To avoid over-exposure of the bands in Complex II, we reduced the exposure time, which may result in under-exposure of other bands and conceal the differences. A more accurate detection of WB bands in Fig.7B by grayscale analysis revealed that knocking out Ets1 significantly increased all the five complex bands by ~3-4 folds (below, B).

(b) As for Fig.7F, though grayscale analysis suggested that Complex I, III, and IV have statistically differences (below, C), we agree with the reviewer that the effect of M2 CDM are very weak. This may be attributed to the relatively short timing of M2 CDM treatment in this *in vitro* assay: the beige differentiated SVF cell (at day 5) were treated with/without M2 CDM for only 24h.

To further clarify the biological function of adipocyte Ets1, as well as M2 macrophages stimuli, in regulating beige adipose mitochondria accumulation *in vivo*, we recently performed M2 macrophage adoption assay, either on control or Ets1 knock-in mice. Our results suggested that while M2 macrophage adoption raised the mitochondria complex content in control littermates, Ets1 knocking in significantly reversed this effect (below, D).

In the revised manuscript, we added the m2 adoption assay in Fig.7, and removed the m2 CDM co-culture assay into supplementary Figure.

Figure above:

Fig.7A-B: Western blot analysis of mitochondrial complex in mature adipocytes derived from *in vivo* WAT of Ets1 adipocyte knock-in (Fig.7A) and knock-out mice (EAKO) (Fig.7B), in comparison with corresponding control littermates.

Fig.7D: Western blot analysis of the level of mitochondria complex and Ets1 in SVF cells with/without M2 CDM coculturing.

New data A: Grayscale analysis of Fig.7A.

New data B: Grayscale analysis of Fig.7B.

New data C: Grayscale analysis of Fig.7F.

New data D: Western blot analysis of the level of mitochondria complex in iWAT of mice. M2 BMDMs were adopted into each side of iWAT, at day 1, 3 and 5 respectively, ensuring that the amount of each injection was 10^6 . The control group was injected with the same volume of PBS. The mice were then given 6-OHDA injections at day 5, 7, and 9 to operate sympathectomy. At day 10, mice were cold stressed at 10 °C for 6 h and then 4 °C for 4 days, after which samples were collected. N=6 per group.

9. The data in Figure 3A is very dramatic, but one wonders what the survival of EA^{+/-} mice or EAKO mice would be?

Response:

Thank you for the question. The heterozygous *ki/-*cre⁺* mice, together with their control littermates, were able to survive the complete 7-days cold stimulation.

The survival data was recorded under a strictly controlled cold exposure process: mice were moved from SPF animal rooms into a 10 °C ventilated freezer for adaption. Six hours later, the freezer was switch to 4 °C to officially start cold stimulation. Under this standard process, all homozygous knock-in mice died (n=7, Fig.3A), while heterozygous mice were able to survive (n=5) until the end of the experiment and complete data collection (Fig.3B). Both knock-in and knock-out mice were repeated at least two times and obtained similar survival rate.

As for *Ets1* KO mice, we currently do not have enough age and gender-matched mice for this assay. Based on our result that blocking *Ets1* raised the mitochondria content (Fig.7B), it is reasonable to speculate that *Ets1* KO mice should have better resistance to cold stress, at least not worse.

Figure above:

Fig.3A: Daily survival rate of EC^{+/+} and EA^{+/+} mice after a 4 °C cold stress.

Fig.3B: Rectal temperature of EC^{+/-} and EA^{+/-} mice after a 4 °C cold stress.

Fig.7B: Western blot analysis of mitochondrial complex in mature adipocytes derived from in iWAT of EAKO and control littermate.

10. The internal structure of the mitochondrial cristae seems significantly altered in Figure 7C. One wonders if a protein such as TMEM11, which controls cristae length and orientation and is involved in mitophagy, might be altered in expression (see PMID: 36795401).

Response:

This is a great question! We performed corresponding assay and our data suggested that Ets1 negatively regulate the expression of TMEM11, however, the regulatory effect of Ets1 on mitochondria morphology can hardly be attributed to TMEM11.

We also noticed the significant decrease in the length of mitochondrial cristae in Ets1 knock-in adipocytes (Fig.7C). However, no significant changes were found in the expression of Drp1/Mfn (Fig.6H).

Figure above: Fig. 7C: Transmission electron microscopy showing the structure of mitochondria. iWAT samples were separated from EC+/+ and EA+/+ mice, after 7 days of CL316243 treatment.

Fig. 6H: qPCR testing of the mRNA levels of mitochondrial dynamic genes in mature adipocytes derived from iWAT of EC+/+ and EA+/+ mice.

The mitochondria protein TMEM11 regulate mitochondrial morphogenesis. Blocking TMEM11 significantly elongated the cristae membranes (PMID: 36795401), and the PMI (drosophila TMEM11 homolog) mutant flies revealed similar phenotypes (PMID: 23264743). Interestingly, the impact of TMEM11 on mitochondrial morphogenesis are independent of DRP1/MFN (PMID: 21274005).

Therefore, we analyzed whether Ets1 affects mitochondrial morphology by regulating TMEM11. Both the immunoblot (A, C, below) and qPCR assay (B, D, below) suggested that the expression of TMEM11 were negatively associated with Ets1 (A-D, below). But it seems that this conflicts with the phenotype of Ets1: in Ets1 knock-in adipocytes, the expression of TMEM11 is significantly reduced, and theoretically, the mitochondrial cristae should become longer at this time. However, in fact, the cristae of Ets1 knock-in adipocytes significantly shortened, indicating that the regulatory effect of Ets1 on mitochondria morphology can hardly be attributed to TMEM11.

Figure above: Western blot (A, C) and qPCR (B, D) analysis of TMEM11 in mature adipocytes derived from iWAT of Ets1 adipocyte knock-out (A, B) and knock-in mice (C, D), in comparison with corresponding control littermates.

Reviewer #3 (Remarks to the Author):

Wu et al. reported a set of data associated with the role of Est1 in mediating the adipose tissue beige process in response to cold stress. Several questions and concerns as follow could challenge the conclusions of this study:

1) Body weight and fat weight after 6-OHDA treatment should be provided.

Response:

Thank you for the question. In our research, the 6OHDA treatment had minimal effect on either body or fat weight.

As the referee's suggested, we added corresponding control groups in the revision and results showed that compared to the sham group, injection of 6OHDA or clodronate liposome, or both, had no significant effect on the body weight or fat weight (below, added in supp-Fig.1).

Figure above: Body weight (A) and Fat weight (B) after 6OHDA treatment.

2) In Fig 1E-H, the control group without 6OHDA and clodronate liposome treatment should be included.

3) The data in Fig 1 are not sufficient to support the conclusion: "macrophages alone could induce adipose beigeing independent of SNS input" (lines 109-110, page 6). In Fig 1, a group treated with only clodronate liposomes (without 6OHDA) should be included to demonstrate the roles of macrophages.

Response to 2) and 3):

Thank you for the comment.

We noticed that both you and Reviewer 2 are concerned about the lack of corresponding control groups in Fig.1. The assay was then repeated with four groups: (1) sham group, (2) clodronate liposome (Clod. Lipo.) injection group, (3) 6OHDA injection group, and (4) 6OHDA+Clod Lipo group.

Compared with control group without 6OHDA or clodronate liposome treatment, both the macrophage depletion and SNS ablation can significantly inhibit adipocyte beigeing and thermogenesis, manifested as lower body temperature (below, A), less expression of thermogenic genes (below, B-C), and corresponding morphological changes after cold stimulation (below, D). Although the effect of SNS seems more intense than the effect of macrophage, it is worth noticing that the effect of SNS ablation

alone was still much smaller than SNS ablation plus macrophage depletion (group 4), indicating macrophages can independently promote adipose beiging.

Figure above:
A-D: Wild type C57/BL6j mice received intraperitoneal injection of 6-OHDA (100 mg/kg) or vehical, on day-5, -3, and -1. At day 0, mice were subcutaneously injected 4 times (2 for each side) with 50 ul clodronate liposomes or control liposomes into iWAT. The liposomes injection was repeated once at day 2. All the four group of sham (n=3), 6OHDA alone (n=3), Clod Lipo alone (n=3), and 6OHDA+Clod Lipo (n=3) were cold stressed for 4 days, started from Day 0.

A: Rectal temperature measured after a 4-day cold stress.
B: Western blot analyzing the protein level of thermogenic marker Pgc1α and Ucp1.
C: Quantitative PCR analyzing the mRNA level of beige adipose marker genes.
D: General morphology of iWAT (up) and H&E staining of iWAT (down).

4) Wrong labels: should be Fig 1J on line 121, Fig 1K on line 123, and Fig 1L on lines 128 and 131.

Response:

Sorry for the mistake, changed as the referee's suggestions.

5) What could be the mechanisms underlying the larger adipocyte size in 6OHDA+CL (Fig 1H)?

Response:

Thank you for the question. The reviewer's mentioned larger adipocyte size, as in this question and in question 10 and 12, is not quite accurate. We measured cell size through two methods, WGA staining, as well as genomic DNA content measurement, and both assays suggested no significant changes in cell size.

When observing adipose tissue via H&E staining (Fig 1H), compared to fat with low thermogenic capacity, adipocyte with high heat production capacity contains smaller lipid droplet and thus become multilocular. This multilocular adipocyte could be easily mistaken as a subset of small cell, thus H&E staining is not suitable for measuring the size of adipocyte.

We then measured cell size via two methods.

1) Staining the tissue section with wheat germ agglutinin (WGA), a member of the lectin family that binds to N-acetyl-D-glucosamine and sialic acid residues found on the surface of cell membranes, could highlight the cell surface membranes. Immunofluorescence staining of WGA suggested that compared to the 6OHDA treated adipose, the 6OHDA+Clod Lipo group had similar cell size (below, A).

2) Extracting genomic DNA from adipose tissue (~50mg), measuring the DNA content and dividing it by the weight of the used sample. The obtained ng DNA/mg tissue can roughly reflect the number of nuclei per unit mass of adipose tissue, and can also roughly reflect cell size: samples with lower ratios have larger cell size. Consistent with the WGA staining, DNA ratio of the two groups was also comparable (below, B).

Figure above: IWAT of Wild type C57/BL6j mice pre-received intraperitoneal injection of 6-OHDA were analyzed.

Fig.1H: H&E staining of iWAT section.

New data A: WGA staining of iWAT section.

New data B: Calculated genome DNA content per mg iWAT tissue.

Therefore, no significant changes in the size of adipocytes were found in the analysis of cell membrane and genomic DNA, and the larger adipocyte size revealed by H&E staining could be visual illusion.

6) Ets1 is among the DBPs in Fig 1K(a)? From the ATAC-seq data, could any genes be regulated by Ets1(b)? consistent with the findings from ChIPseq analysis(c)?

Response:

Thank you for the question.

(a) Yes, Ets1 is among the DBPs in Fig. 1K. The specific names of the 51 DBPs in Fig.1K are shown in Fig.1L, and Ets1 is one of the 29 intersecting DBPs, which may have physical functions in both white and beige adipocytes.

(b-c) We screened 4 candidates, with 3 could be regulated by Ets1 and one that not likely be regulated by Ets1, and the ChIP-seq analysis showed consistent result.

As the review's suggested, we profiled the potential transcription binding site of Ets1 on the promoter region (-1000~+100) of all the other 50 DBPs, via JASPAR database. We selected three genes that contains multiple potential binding sites (ERF, Gabpa, Sp3), and one gene without any potential binding site (Klf14) (Panel below, A). Chip-seq analysis showed consistent result: Ets1 could binds to the promoter regions of Erf, Gabpa, and Sp3, but not Klf14 (Panel below, B). Quantitative PCR revealed suppressive effect of Ets1 on Erf, Gabpa, and Sp3, while the Klf14 were not affected (Panel below, C-D). Therefore, Ets1, a classic transcription factor, can regulate the expression of part of the 51 DBPs that discovered by ATAC-seq.

Figure above:

A: JASPAR database analyzing the potential Ets1 binding site on the promoter (-999~+100) region of the candidate genes. Red box: Ets1 binding site.

B: The IGV tool was used to visualize the binding peaks of Ets1 on the promoter region of the candidate genes. Transcription initiation sites and directions marked by black triangles.

C: Quantitative PCR testing of the mRNA level of candidate genes, in 3T3-L1 cells pre-infected with/without Ets1 overexpression adenovirus.

D: Quantitative PCR testing of the mRNA level of candidate genes, in 3T3-L1 cells pre-infected with/without Ets1 knock-down adenovirus.

7) The connection of sequencing data in Fig 1J and published dataset in Fig 1L and 1M&N is not well established, especially in demonstrating the interaction between macrophages and adipocytes.

Response:

Thank you for the suggestion. We rewrote the mentioned part as follow and hope that this will make it clear.

... ..

To explore the mechanism through which macrophage directly modulate adipocyte being, we set up an analysis pipeline (Fig. 1E). Considering the key role of transcriptional regulation in the differentiation of beige adipocytes [33], we focused on DNA binding proteins (DBPs), such as transcription factors and epigenetic regulators. The candidate DBPs were screened according to the following limits.

(1) The DBPs is bio-active in adipocyte lineage. We first performed the Assay for Transposase Accessible Chromatin with high-throughput sequencing (ATAC-seq) to profile the global open chromatin structure of the white and beige adipocytes. Comparison with a published FAIRE-seq dataset [34] revealed that our ATAC-seq had more reliable sequencing signals with much lower background (Supp-Fig. 1E). Motif analysis of the open chromatin regions, which can identify active and functional DBPs, revealed 30 active DBPs in white adipocytes and 50 in beige adipocytes, with 29 shared DBPs (Fig. 1G).

(2) The adipocyte bio-active DBPs could response to M2 macrophage stimuli. Using a published RNA-seq dataset for macrophage co-cultured adipocytes (GSE228094), we found that compared with the mono-cultured adipocytes, the adipocytes co-cultured with M2 macrophages showed reduced expression of *Ets1*, and increased *Klf4* expression (Fig. 2K). *Klf4* is considered an essential early regulator of adipocyte differentiation [35], but no clear physiological function has been established for *Ets1* in adipocytes. Interestingly, unlike *Klf4*, adipocytes co-cultured with M1 macrophages (Fig. 1G), which promote inflammation and impede beige adipogenesis [36], showed increased *Ets1* expression. These results suggested that *Ets1* could be a potential regulator of macrophage–adipocyte communication.

(3) The adipocyte bio-active, M2 macrophage responsible DBP, should be highly associated with the beige adipogenesis process, either in its expression or in its activity. Using a published adipose tissue sc-sequencing database [8], we profiled the expression of *Ets1* in a variety of cells located in adipose tissue and found that mice fed a normal chow diet (NCD) showed *Ets1* expression mainly in ASPCs, B cells, macrophages, endothelial cells, and mesothelial cells (Fig.1H). By contrast, only a very small portion of the mature adipocytes showed *Ets1* expression (Fig.1H). Interestingly, mice fed a high-fat diet (HFD) had higher numbers of mature adipocytes that expressed *Ets1*

(Supp-Fig. 1F). We then conducted a trajectory analysis of ASPCs and mature adipocytes to determine the developmental pathway (Supp-Fig. 1G-H). Differential expression analysis identified several gene cohorts that changed during this ASPCs-mature adipocytes transition. Notably, the transition was marked by downregulation of Ets1, together with other gene signatures found in (but not specific to) ASPCs, such as Dpp4, Fstl1 [37], and Egfr [38], and upregulation of mature adipocyte marker genes, such as Pparg, Pnpla3, Lpl, Adipoq, and Fabp4 (Fig. 1I). These data revealed a negative correlation between Ets1 and the differentiation process from ASPCs to adipocytes.

... ..

8) Ets1 data in Fig 2C and 2D are not consistent. Ets1 expression in mature adipocytes (Fig 1M and Fig 2C and 2F) is very low or barely detectable, but Ets1 expression of 0day mature adipocyte-derived from iWAT in Fig 2D is robust.

Response:

Thank you for the keen observation and discovering the problem of this figure. After repeated checking, we found that the annotation of this panel was incorrect and mixed up with Suppl-Fig.3A (also see response to 13).

In this study, both whole fat pad and fat pad derived mature adipocyte were widely used. We discovered that due to communication errors between the first author and the corresponding author, error occurred in the labeling of the following two assays. In Fig.2D, the fat pad was actually used instead of the previously mentioned mature adipocyte; while in suppl-Fig.3A, the mature adipocytes were analyzed, rather not fat pad. We are terribly sorry for the mismatch label of these two panels.

In order to eliminate the concerns of the reviewer and also confirm the previous data, we conducted a more detailed study on Fig.2D and data suggested that Ets1 was indeed expressed in mature adipocyte (below, A).

Figure Above: Western blot showing the protein levels of Ets1 in iWAT fat pad (Fig.2D), in mature adipocyte (A) or in SVF (B) derived from iWAT of mice pretreated with cold stress for indicate time.

In addition, we would like to point out that although the expression of Ets1 in adipocytes is significantly lower than that in SVF cells, it is still expressed and detectable, thus may not refer it as “barely detectable”. For example, in Fig.2C, as exposure increase, Ets1 expression can also be detected in mature adipocytes (Below,

A). Besides, our data in Fig.2E, and Suppl-Fig.1B (below) also suggested that Ets1 is low expressed in mature adipocytes, but not completely absent.

Figure Above:

Fig. 2C: Western blot of Ets1 in mature adipocytes (adipo) and the SVF cells of iWAT and eWAT (n=2).

Fig. 2E: SVFs separated from wild type C57 were differentiated into beige adipocytes. At indicated time point after differentiation, protein level of Ets1 were analyzed by western blot (n=2).

Supp-Fig. 1B: UMAP projection of only the ASPC and mature adipocytes of both chow diet and high fat diet fed mice (left). The expression of Ets1 across these cells were profiled (right). A published single cell transcription sequencing dataset (GSE176171) were used.

New data A: long exposure image of Fig.2C.

We sincerely apologize again for the mixed usage of the annotations in Supp-Fig.3A and Fig.2D of the original manuscript, as well as the possible misleading of the endogenous expression level of Ets1 in Fig.2C, and thank you for your efforts in improving this work.

9) To mimic in vivo conditions, authors should use CDM from adipose tissue macrophages for coculturing assays.

Response:

Thank you for the comment. We separated the adipose tissue macrophage (ATM) and performed the conditional medium (CDM) coculturing assay, and the result were basically the same as BMDM CDM coculturing assay.

A Cell paper published at 2014 (PMID: 24906148) discovered that 5°C cold stress progressively increased macrophage content of iWAT, and this newly recruited macrophages is necessary for beige adipogenesis. To fully mimic the in vivo conditions, the wild type C57/BL6j mice were cold stressed for 48h, the F4/80⁺, CD11b⁺ ATMs from 40 mice were sorted via flow cytometry (panel A, below), collected and cultured in dishes to harvest ATM CDM. The following coculturing assay suggested that ATM CDM significantly activates thermogenic marker gene, raised mitochondria complex content, and enhanced oxygen consumption rate, whereas Ets1 overexpression completely reversed these effects (panel B-D, below).

The above data has been added in Fig.7 in the revised manuscript.

Figure above:
 A: Adipose tissue macrophage (ATM) separation. Wild type C57/BL6j mice were cold stressed for 48h, the F4/80⁺, CD11b⁺ ATMs from both iWAT and eWAT were sorted via flow cytometry, collected and cultured in dishes to harvest ATM CDM.
 B-D: SVFs separated from iWAT of wild type C57 mice were cultured in vitro. The plate-fully-covered cells were infected with indicate adenovirus for 6h, then induced differentiation towards beige adipocytes. 5 days later, the cells were treated with 50% ATM CDM for 24h.
 B: Western blot analysis of the level of OXPHOS, Ets1, and Ucp1 (n=2).
 C: Quantitative PCR showing the mRNA levels of Ets1 and thermogenic markers (n=4).
 D: Oxygen consumption rate (OCR) measured at Day 8 (n=6).

10) What could be the mechanisms underlying the larger adipocyte size in EA+/- mice (Fig 3C)?

Response:

Thank you for the question. As mentioned in previous response, the larger adipocyte could be visual illusion caused by H&E staining (Fig.3C), which is unsuitable for comparing cell size.

We performed WGA staining (below, A), as well as genome DNA content measurement (below, B), results suggested that the adipocyte size is comparable between the mentioned groups.

Figure above: IWAT of EC+/- and EA+/- mice pretreated with 7-day cold stress were analyzed.
 Fig.3C: H&E staining of iWAT and BAT section.
 New data A: WGA staining of iWAT and BAT section.
 New data B: Calculated genome DNA content per mg iWAT and BAT tissue.

11) Body weight and fat weight after overexpression of Ets1 should be presented.

Response:

Thank you for the question. The body weight (Below, A) and fat weight (Below, B-C) were recorded and data suggested that homozygous adipocyte Ets1 knock-in mice had similar body weight and fat weight with control mates. These data have been added in the supp-Fig.3 in the revised manuscript.

Figure above:
Body weight (A) and fat weight (B-C) of homozygous Ets1 knock-in mice and control littermate.

12) The adipocyte size phenotypes shown in Fig 3C and 3F occurred in response to cold stress? or these phenotypes were formed because of Ets1 overexpression, even housed at room temperature.

Response:

Thank you for the question. As mentioned in response to question 5 and question 10, the larger adipocyte could be visual illusion caused by H&E staining (Fig.3C), which is unsuitable for comparing cell size.

Like the heterozygous Ets1 knock-in mice data shown in response 10 (Fig.3C), we also performed WGA staining (below, A), as well as genome DNA content measurement (below, B), using fat pad separated from homozygous Ets1 knock-in mice. Results suggested that the adipocyte size is comparable between the mentioned groups.

Figure above: iWAT of EC+/+ and EA+/+ mice pre-treated with CL316243 were analyzed.
Fig.3F: H&E staining of iWAT and BAT section.
New data A: WGA staining of iWAT and BAT section.
New data B: Calculated genome DNA content per mg iWAT and BAT tissue.

13) In Suppl Fig 3A, fat pads were used. Since adipocyte-specific Ets1 knockout and other cells in fat pads, such as macrophages and endothelial cells, are not affected, why Ets1 was completely depleted in EAKO samples?

Response:

Thank you again for discovering the problem of this figure. After repeated examination, we found that the annotation of this image was incorrect, and mix up with Fig.2D (see response to question 8).

In this study, both whole fat pad and fat pad derived mature adipocyte were widely used. We discovered that due to communication errors between the first author and the corresponding author, error occurred in the labeling of the following two assays. In Fig.2D, the fat pad was actually used instead of the previously mentioned mature adipocyte; while in suppl-Fig.3A, the mature adipocytes were analyzed, rather not fat pad. We are terribly sorry for the mismatch label of these two panels.

In order to eliminate the concerns of the reviewer and also confirm the previous data, we conducted the following assay recently. After digestion and centrifugation, the fat pads of knockout mice and control littermate were divided into mature adipocytes and SVF cells. These two parts of cells, as well as the intact fat pads, were subjected to protein extraction and immunoblotting to analyze the expression of Ets1. The WB result revealed that compared with the control mice, the knockout mice contain far more less Ets1, nearly undetectable, in the mature adipocyte, while the Ets1 level in the SVF were comparable (below). In the fat pad lysis samples, Ets1 expression was slightly lower in the KO mice, compared with control rodent (below).

We sincerely apologize again for the mixed use of the annotations in Supp-Fig.3A and Fig.2D of the original manuscript and thank you for your efforts in improving this work.

Figure above: Western blot analysis of Ets1 protein levels in whole fat pads, in mature adipocyte, and in SVF cell.

14) As mature adipocytes from iWAT of lean WT mice expressed a very low abundance of Ets1 and even less in response to cold stress, it is puzzling how ChIP-seq assays were done with these cells. It is possible that other cells such as macrophages or adipocyte precursor cells were bound with adipocytes were co-isolated from iWAT and contributed to Ets1 abundance.

Response:

This is a good question. The probability of the previous ChIP-seq being affected by non-adipocyte was extremely low. To further address the concern of the reviewer, we conducted a ChIP-seq using SVF derived mature beige adipocyte, and the results confirmed the correctness of previous sequencing data.

We believe that the influence of non-adipocyte on the previous sequencing data can be ruled out, for the following reasons:

(1) Although relatively low (compare to SVF), Ets1 is indeed expressed in mature adipocytes (refer to “Response to question 8”). The sample size required for high-throughput sequencing is very low, thus ChIP-seq analysis of Ets1 in mature adipocytes can be achieved.

(2) During sequencing library building, we first collected the mature adipocyte from iWAT, then harvest only the nucleus and washed twice before Cut&Tag. We believed that after these operations, the likelihood of introducing non-adipocytes Ets1 into the sample is very low.

(3) In addition, even if non-adipocyte Ets1 is indeed mixed into the samples, the potential impact can be effectively eliminated during the following differential analysis. Cold stimulation could only affect the expression of Ets1 in adipocyte, while the non-adipocyte cells were unaffected (A-B, below). Hence, if non-adipocytes Ets1 brings sequencing distortion in the cold stimulated mice, it should introduce the same level of sequencing noise in the control mice. Therefore, when comparing these two sets of sequencing data, the potential distortion introduced by non-adipocytes were offset, leaving the differential peaks mainly attributed from adipocyte.

Figure Above: Western blot showing the protein levels of Ets1 in iWAT fat pad (Fig.2D), in mature adipocyte (A) or in SVF (B) derived from iWAT of mice pretreated with cold stress for indicate time.

Considering the above discussion may not fully eliminate the concern raised by the referee, we further conducted Ets1 ChIP-seq using only SVF derived beige adipocytes. Our data suggested that in these *in vitro* differentiated mature adipocytes, Ets1 ChIP-seq also showed binding peaks on the promoter region of autophagy related and mitochondria coding genes (below, A), consistent with the previous data.

Sequencing data of the newly performed SVF derived beige adipocyte-based ChIP-seq has been added to the GSE221335 dataset.

Figure above:
A: The IGV tool was used to visualize the binding peaks of *Ets1* on the promoter region of autophagy-related and mitochondrial genes. RT adipo: adipocyte separated from iWAT of mice kept at room temperature were used; CS adipo: adipocyte separated from iWAT of mice pretreated with cold stress were used; SVF adipo: new data of ChIP-seq using SVF derived mature beige adipocyte.

15) References should be added for lines 274-275.

Response:

Thank you for the comment. Changed according to reviewer's suggestion.

16) How is ASPC isolated for Fig 6A? However, the fig legend stated "SVF", then it should be SVF. Similarly, fig 6F is labeled with ADSC but fig legend states "SVF".

Response:

Thank you for the suggestion. We apologize for the misuse and changed the mentioned "ASPC" to "SVF" in the revised manuscript.

17) All histology or immunofluorescent images should be included with a scale bar.

Response:

Thank you for the comment. Changed according to reviewer's suggestion.

18) What could the factor(s) in M2 CDM regulating *Ets1* expression?

Response:

This is a good question! Our data suggested that the mediator should be small molecule that less than 10KD, but the specific molecule is currently unclear.

We first analyzed the regulatory effects of M1/M2 macrophage derived cytokines on the expression of *Ets1*. Our data suggested that IL-10 (mainly secreted by m2 macrophages), as well as IL-6, IL-1b, and TNF α (mainly secreted by m1 macrophages) treatment could not significantly change the expression of *Ets1*(below, panel A).

Recently, we separated the M2 CDM by using different types of ultrafiltration tubes, which can distinguish the components in the medium by molecular weight, and then treated the SVF cells with these separated components. We found that even with

the smallest ultrafiltration tube, which only allow the passage of molecules less than 10kd, the filtrated solution could still significantly inhibit the expression of Ets1 (below, panel B). This highly suggests that the inhibitory effect of macrophages on Ets1 is unlikely to be exerted through large molecular weight components, such as secreted proteins or exosomes, and the possibility of acting through inflammatory cytokines or chemokines is also relatively small. We speculate that it is possible that certain small metabolites secreted by M2 macrophages inhibit the expression of Ets1 in adipocyte. Further research (such as metabolomics) is still needed, to discover the mediator and clarify the underlying mechanism.

Figure above:

A: Western blot showing the expression of Ets1 in 3T3-L1, which were pretreated with corresponding cytokines for 4h.

B: Western blot showing the expression of Ets1 in SVF cells, which were pretreated with M2 CDM for 4h. Control: 0% M2 CDM; Total: 50% M2 CDM; <10 KD and >10KD: M2 CDM were separated via ultrafiltration tubes, the media containing molecules < 10KD, and > 10KD were collected, and used to treat SVF cells at 50%.

19) While the aim of this study focused on the regulation of M2 macrophages on adipose tissue beiging, there is a limited amount of data supporting how macrophages mediate the adipocyte beiging process (a). In addition, the IL4-induced BMDM (M2 macrophages) are not the same cells as adipose tissue macrophages (b).

Response:

Thank you for the question. Through three independent assays of BMDM CDM co-culture, ATM CDM co-culture, and M2 macrophage adoption, our revised data discovered a pivotal role of adipocyte Ets1 in responding to M2 macrophages stimuli, as well as in promoting beige adipose formation, both *in vitro* and *in vivo*.

(a) To further clarify the biological function of adipocyte Ets1 in responding to M2 macrophages stimuli as well as promoting beige adipose formation *in vivo*, we recently performed macrophage adoption assay, either on control or adipocyte Ets1 knock-in mice. Our results suggested that while M2 macrophage adoption raised the mitochondria content (A, below), promotes the cold induced beige adipogenesis (B, below), up-regulated the body temperature (C, below), enhanced the thermogenic gene expression (D, below), forced Ets1 expression significantly reversed all these effects (A-D, below).

Figure above:

A-D: M2 BMDMs were adopted into each side of iWAT, at day 1, 3 and 5 respectively, ensuring that the amount of each injection was 10^6 . The control group was injected with the same volume of PBS. The mice were then given 6-OHDA injections at day 5, 7, and 9 to operate sympathectomy. At day 10, mice were cold stressed at 10°C for 6 h and then 4°C for 4 days, after which samples were collected. $N=6$ per group.

A: Western blot analysis of the level of OXPHOS, Ets1, and Ucp1.

B: General morphology of iWAT (left) and H&E staining of iWAT (right).

C: Rectal temperature measured during cold stress.

D: qPCR testing Ets1, macrophage marker F4/80, and thermogenic fat marker genes.

(b) As for the “BMDM are not the same cells as adipose tissue macrophages”, we separated ATM via flow cytometry and performed the conditional medium (CDM) co-culture assay as the reviewer suggested in question 9. The data were consistent with the BMDM-based assay, the detailed data and discussion are shown in “response to question 9”.

REVIEWERS' COMMENTS

Reviewer #1 (Remarks to the Author):

The authors have addressed my concerns.

Reviewer #2 (Remarks to the Author):

The authors have done a very thorough and thoughtful revision of their manuscript to address reviewer comments. They have addressed all of my points adequately and I feel the manuscript is ready for publication.

Reviewer #3 (Remarks to the Author):

no additional questions/comments